# Modeling the *Trichodesmium* sp. related N₂ fixation : driving processes and impacts on primary production in the tropical Pacific Ocean

Dutheil Cyril[1,2], Aumont Olivier[2], Gorguès Thomas[3], Lorrain Anne[4], Bonnet Sophie[5], Rodier Martine[6], Dupouy Cécile[1,5], Shiozaki Takuhei[7], and Menkes Christophe[1,2]

[1]Centre IRD, Nouméa, New Caledonia
[2]LOCEAN Laboratory, IPSL, Sorbonne Universités (UPMC, Univ Paris 06)-CNRS-IRD-MNHN, Paris, France
[3]Laboratoire d'Océanographie Physique et Spatiale (LOPS), Univ. Brest-CNRS-Ifremer-IRD, Plouzané, France
[4]LEMAR, UMR 6539, UBO-CNRS-Ifremer-IRD, IUEM, Plouzané, France
[5]Aix Marseille Université, CNRS/INSU, Université de Toulon, IRD, Mediterranean Institute of Oceanography (MIO) UM 110, 13288, Marseille, France
[6]Environnement Insulaire Océanien (EIO), UMR 241(Univ. de Polynésie Française, IRD, ILM, IFREMER), Tahiti, French Polynesia
[7]Research and Development Center for Global Change, Japan Agency for Marine-Earth Science and Technology, Yokosuka, Japan

**Correspondence:** Cyril Dutheil (cyril.dutheil@ird.fr)

**Abstract.** Dinitrogen fixation is now recognized as one of the major sources of bio-available nitrogen in the ocean. Thus,
N₂ fixation sustains a significant part of the global primary production by supplying the most common limiting nutrient for
phytoplankton growth. The «Oligotrophy to UlTra-oligotrophy PACific Experiment» (OUTPACE) improved the data coverage
of the Western Tropical South Pacific, an area recently recognized as a hotspot of N₂ fixation. This new development leads
us to develop and test an explicit N₂ fixation formulation based on the *Trichodesmium* physiology (the most studied nitrogen
fixer) within a 3D coupled dynamical-biogeochemical model (ROMS-PISCES). We performed a climatological numerical
simulation that is able to reproduce the main physical (e.g Sea Surface Temperature) and biogeochemical patterns (nutrients,
and chlorophyll concentrations as well as N₂ fixation) in the tropical Pacific. This simulation displayed a *Trichodesmium*
regional distribution that extends from 150°E to 120°W in the south tropical Pacific, and from 120°E to 140°W in the north
tropical Pacific. The local simulated maximums were found around islands (Hawaii, Fiji, Samoa, New Caledonia, Vanuatu).
We assessed that 15% of the total primary production may be due to *Trichodesmium* in the Low Nutrient, Low Chlorophyll
regions (LNLC) of the Tropical Pacific. Comparison between our explicit and the often used (in biogeochemical models)
implicit parameterization of N₂ fixation showed that the latter leads to an underestimation of N₂ fixation rates by about 25% in
LNLC regions. Finally, we established that iron fluxes from island sediments control the spatial distribution of *Trichodesmium*
biomasses in the western tropical south Pacific. Noteworthy, this last result does not take into account the iron supply from
rivers and hydrothermal sources, which may well be of importance in a region known for its strong precipitation rates and
volcanic activity.

# 1  Introduction

Nitrogen is known to be the most common limiting nutrient for phytoplankton growth in the modern world ocean (Moore et al., 2013), especially in the Low Nutrient, Low Chlorophyll (LNLC) ecosystems (Arrigo, 2005; Gruber, 2005). Characterizing the processes governing nitrogen sources and sinks to and from the ocean is therefore central to understanding oceanic production, organic matter export and food web structure. Atmospheric dinitrogen ($N_2$) dissolved in seawater is by far the dominant form of N present in the ocean, i.e. the $N_2:NO_3^-$ ratio typically exceeds 100 in surface waters. However, most phytoplankton species cannot assimilate $N_2$, and grow using reactive forms of nitrogen such as nitrate, ammonium and dissolved organic compounds. Some planktonic prokaryotic microorganisms, called "diazotrophs" use an enzyme, the nitrogenase, to fix $N_2$ and convert it into ammonia ($NH_3$) and ultimately ammonium ($NH_4^+$). At the global scale, they provide the major external source of reactive nitrogen to the ocean (Gruber, 2008), and support up to 50% of new production in tropical and subtropical (LNLC) regions (Bonnet et al., 2009; Capone, 1997; Deutsch et al., 2007b; Karl et al., 1997; Moutin et al., 2008; Raimbault and Garcia, 2008). These organisms are physiologically and taxonomically diverse including cyanobacteria, bacteria and archaea (Zehr and Bombar, 2015; Delmont et al., 2017).

Autotrophic diazotrophs have been far more intensively studied than heterotrophic diazotrophs, whose contribution to global $N_2$ fixation remains unclear (Turk-Kubo et al., 2014; Bombar et al., 2016; Moisander et al., 2017). Autotrophic diazotrophs have been characterized both in the field and through laboratory experiments and their physiology is consequently better known (Bergman et al., 2013; Küpper et al., 2008; Mulholland and Capone, 2001, 2000; Ohki et al., 1992; Ramamurthy and Krishnamurthy, 1967; Rubin et al., 2011). Cyanobacterial (autotrophic) diazotrophs are composed of 3 main groups: 1) the filamentous diazotrophs including the colonial, non-heterocyst-forming *Trichodesmium*, 2) the heterocyst-forming symbionts associated with diatoms (DDAs), and 3) the unicellular cyanobacterial diazotrophs (UCYN, phylogeneticaly divided into three groups : UCYN-A, B, and C). It has been established that autotrophic diazotrophs growth rates are typically one order of magnitude lower than those of non-diazotrophs (Breitbarth et al., 2008; Falcón et al., 2005; Goebel et al., 2008; LaRoche and Breitbarth, 2005). This can be related to the high energetic demand (Postgate, 1982) required to convert $N_2$ to $NH_3$ as compared to that necessary to assimilate nitrate or ammonia. This low growth rate (compared to other phytoplankton species) mainly constrains their ecological niches to nitrate-poor regions, where they can be competitive. Moreover, their geographical distribution is constrained by nutrient availability in the photic layer (mainly iron and phosphate) (Berman-Frank, 2001; Bonnet et al., 2009; Mills et al., 2004; Moutin et al., 2005, 2008; Rubin et al., 2011; Rueter, 1988) and temperature (Staal et al., 2003). *Trichodesmium* sp. are present only in water where temperature is above 20°C (Capone, 1997; LaRoche and Breitbarth, 2005; Montoya et al., 2004), while some UCYN can be found in colder and deeper waters (Bonnet et al., 2015; Church et al., 2005; Moisander et al., 2010).

The spatial distribution and rates of $N_2$ fixation have been inferred at the global scale using several tools. Deutsch et al. (2007a) have introduced the tracer P* which represents the excess of P relative to the standard N quota. A decrease in this tracer is then interpreted as $N_2$ fixation, since $N_2$ fixation extracts $PO_4$ alone. More recently, Luo et al. (2014) developed a multiple linear regression that relates $N_2$ fixation from the MAREDAT database (Luo et al., 2012) to environmental conditions (nutrients, SST,

irradiance, MLD,...) in order to build a statistical model for global N$_2$ fixation distribution.
Numerical models have also been used as they allow to overcome the scarcity of observations that may limit the implementation
of the two previous approaches (Aumont et al., 2015; Bissett et al., 1999; Dutkiewicz et al., 2012; Keith Moore et al., 2006;
Krishnamurthy et al., 2009; Monteiro et al., 2011; Moore et al., 2013; Tagliabue et al., 2008). They can notably be used to
investigate the spatial and temporal variability of dinitrogen fixation and to study its controlling environmental factors. In these
models, N$_2$ fixation has been implemented in various ways. Some models use implicit parameterizations (Bissett et al., 1999;
Maier-Reimer et al., 2005; Assmann et al., 2009; Aumont et al., 2015) to derive N$_2$ fixation from environmental conditions
(mainly nitrate, phosphate and iron concentrations, temperature and light) without explicitly simulating any nitrogen fixing
organisms. Alternatively, other models rely on the explicit descriptions of diazotrophs (Moore et al., 2004; Dunne et al., 2013)
that have mainly been developed from the knowledge derived from laboratory experiments focused on *Trichodesmium* sp.
(Fennel et al., 2001; Hood et al., 2001; Moore et al., 2001). Noticeably, several modeling studies have been especially focused
on the role of iron in controlling the distribution of diazotrophs and N$_2$ fixation (Keith Moore et al., 2006; Krishnamurthy
et al., 2009; Moore et al., 2004; Tagliabue et al., 2008). Indeed, a realistic representation of marine iron concentrations has
been stressed as a key factor to adequately simulate the habitat of diazotrophs (Monteiro et al., 2011; Dutkiewicz et al., 2012).
Moreover, among the full set of studies focusing on the spatial distribution of N$_2$ fixation, some studies (Berthelot et al., 2017;
Bonnet et al., 2009, 2015; Garcia et al., 2007; Shiozaki et al., 2014) based on oceanographic campaigns have reported high
N$_2$ fixation rates in the Western Tropical South Pacific (WTSP), that has been recently identified as a globally important hot
spot of N$_2$ fixation with rates > 600 μmol N m-2.d-1 (Bonnet et al., 2017). Very high abundances of *Trichodesmium* have
been historically reported in this region (Dupouy et al., 2000, 2011; Moisander et al., 2008; Neveux et al., 2006; Shiozaki
et al., 2014) Stenegren et al., This issue) and have recently been identified as the major contributor to N$_2$ fixation in this region
(Berthelot et al., 2017) Bonnet et al., This issue). However, the reasons for such an ecological success of diazotrophs in this
region are still poorly understood.
In this study, we aim at bringing new insights on this known, but poorly understood, "N$_2$ fixation hotspot". This study ambitions
to understand the spatial and temporal distribution (i.e. seasonal variability) of *Trichodesmium* and to evaluate the potential
impact of *Trichodesmium* fixers on the biogeochemical conditions of the WTSP. We will specifically address the following
overarching questions: (i) What are the mechanisms that structure the *Trichodesmium* distribution in the WTSP, particularly
around the South West Pacific islands, and (ii) what is the biogeochemical impact of N$_2$ fixation in this region? Noteworthy,
this study is also taking advantage of the sampling done during the « Oligotrophy to UlTra-oligotrophy PACific Experiment »
(OUTPACE), which nicely complement the data coverage in the south west Pacific, and allow a better characterization of the
processes responsible for the spatial and seasonal variability of the N$_2$ fixation.
To fulfill our objectives, we have implemented an explicit representation of the nitrogen fixers, based on the *Trichodesmium*
physiology, in a biogeochemical model. The first section of this study describes the experimental design and the observations
used in our study, while the second part of the paper provides a validation of our reference simulation with an analysis of the
*Trichodesmium* compartment and its impacts on the biogeochemical conditions of the Tropical Pacific. In the discussion, the
impact of iron from islands sediment on dinitrogen fixation is considered as well as the added value of an explicit dinitrogen
fixer compartment rather than a simpler implicit representation of $N_2$ fixation. Finally, implications for and limits of our
modeling exercise are detailed in the conclusion.

## 2   Methods

### 2.1   Coupled dynamical (ROMS)-biogeochemical (PISCES) model

#### 2.1.1   ROMS

In this study, we used a coupled dynamical-biogeochemical framework based on the regional ocean dynamical model ROMS
(Regional Oceanic Modeling System, (Shchepetkin and McWilliams, 2005) and a new version of biogeochemical model
PISCES (Pelagic Interactions Scheme for Carbon and Ecosystem Studies). The ocean model configuration is based on the
ROMS-AGRIF (Penven et al., 2006) computer code and covers the tropical Pacific region [33°S-33°N;110°E-90°W]. It has 41
terrain-following vertical levels with 2-5 m vertical resolution in the top 50 meters of the water column, then 10-20 m resolution
in the thermocline and 200-1000 m resolution in the deep ocean. The horizontal resolution is 1°. The turbulent vertical mixing
parameterization is based on the non-local K profile parameterization (KPP) of Large et al. (1994). Open boundaries condi-
tions are treated using a mixed active/passive scheme (Marchesiello et al., 2001). This scheme is used to force our regional
configuration with monthly climatological large-scale boundary conditions from a ½° ORCA global ocean simulation (details
available in Kessler and Gourdeau (2007)), while allowing anomalies to radiate out of the domain. The use of similar ROMS
configurations (e.g vertical resolution, mixed active/passive scheme, turbulent vertical mixing parameterization) in the WTSP
is largely validated through studies demonstrating skills in simulating both the surface (Jullien et al., 2012, 2014; Marchesiello
et al., 2010) and subsurface ocean circulation (Couvelard et al., 2008).
To compute the momentum and fresh water/heat fluxes, we also use a climatological forcing strategy. Indeed, documenting
the inter-annual to decadal variability is beyond the scopes of our study, which justifies using climatological forcing fields.
A monthly climatology of the momentum forcing is computed from the 1993-2013 period of the ERS1-2 scatterometer stress
(http://cersat.ifremer.fr/oceanography-from-space/our-domains-of-research/air-sea-interaction/ers-ami-wind). Indeed, ERS de-
rived forcing has been shown to produce adequate simulations of the Pacific Ocean dynamics (e.g. Cravatte et al., 2007). A
monthly climatology at 1/2° resolution computed from the Comprehensive Ocean–Atmosphere Data Set (COADS; Da Silva
et al. (1994)) is used for heat and fresh water forcing. In our set-up, ROMS also forces on line a biogeochemical model using a
WENO5 advection scheme (i.e. five order weighted essentially non-oscillatory scheme; Shchepetkin and McWilliams, 1998).
After a one year spin-up we stored 1-day averaged outputs for analysis.

#### 2.1.2   PISCES

In this study, we use a quota version of the standard PISCES model (Aumont and Bopp, 2006; Aumont et al., 2015), which
simulates the marine biological productivity and the biogeochemical cycles of carbon and the main nutrients (P, N, Si, Fe).
This modified model, called PISCES-QUOTA, is extensively described in Kwiatkowski et al. (2018). Our version is essentially
identical to Kwiatkowski's version that included an added picophytoplankton group, except that this latter group has been
removed and replaced by a *Trichodesmium* compartment. Here we only highlight the main characteristics of the model and
the specifics of our model version. Our version of PISCES-QUOTA has then 39 prognostic compartments. As in the standard
PISCES version, phytoplankton growth is limited by the availability in five nutrients: nitrate and ammonium, phosphate, sil-
icate and iron. Five living compartments are represented: three phytoplankton groups corresponding to nanophytoplankton,
diatoms, and *Trichodesmium* and two zooplankton size-classes that are microzooplankton and mesozooplankton. The elemen-
tal composition of phytoplankton and non-living organic matter is variable and is prognostically predicted by the model. On
the other hand, zooplankton are assumed to be strictly homeostatic, i.e. their stoichiometry is kept constant (e.g. Meunier et al.,
2014; Sterner and Elser, 2002). Nutrients uptake and assimilation as well as limitation of growth rate are modeled according to
the chain model of Pahlow and Oschlies (2009). The P quota limits N assimilation which in turns limits phytoplankton growth.
The phosphorus to nitrogen ratios of phytoplankton are described based on the potential allocation between P-rich biosynthesis
machinery, N-rich light harvesting apparatus, a nutrient uptake component, the carbon stores, and the remainder (Daines et al.,
2014; Klausmeier et al., 2004). This allocation depends on the cell size and on the environmental conditions.
Nutrients are delivered to the ocean through dust deposition, river runoff and mobilization from the sediment. The atmospheric
deposition of iron is derived from a climatological dust simulation (Tegen and Fung, 1995). The iron from sediment is recog-
nized as a significant source (Johnson et al., 1999; Moore et al., 2004). This iron source is indeed parameterized in PISCES as,
basically, a time-constant flux of dissolved iron (2 $\mu$mol.m$^{-2}$.day$^{-1}$) applied over the whole sediment surface and modulated
depending only on depth. A detailed description of this sedimentary source is presented in Aumont et al. (2015). The initial
conditions and biogeochemical fluxes (iron, phosphorus, nitrate, ...) at the boundaries of our domain are extracted from the
World Ocean Atlas 2009 (https://www.nodc.noaa.gov/OC5/WOA09/woa09data.html).

### 138 2.1.3 *Trichodesmium* compartment

For the purpose of this study, we implemented in the PISCES-QUOTA version an explicit representation of *Trichodesmium*.
Therefore, as already stated, five living compartments are modeled with three phytoplankton groups (nanophytoplankton,
diatoms, and *Trichodesmium*) and two zooplankton groups (microzooplankton, and mesozooplankton). Similarly to nanophy-
toplankton, the equation of *Trichodesmium* evolution is computed as follows:

$$\frac{\partial Tri_C}{\partial t} = (1 - \delta^{Tri})\mu^{Tri}Tri - \zeta^{Tri}_{NO_3}V^{Tri}_{NO_3} - \zeta^{Tri}_{NH_4}V^{Tri}_{NH_4} - m^{Tri}\frac{Tri_C}{K_m + Tri_C}Tri_C$$

$$- sh \times \omega^{Tri}P^2 - g^Z(Tri)Z - g^M(Tri)M \tag{1}$$

In this equation, $Tri_C$ is the carbon *Trichodesmium* biomass, and the seven terms on the right-hand side represent respectively
growth, biosynthesis costs based on nitrate and ammonium, mortality, aggregation and grazing by micro- and mesozooplankton.
In our configuration, the photosynthesis growth rate of *Trichodesmium* is limited by light, temperature, phosphorus and iron
availability. Photosynthesis growth rate of *Trichodesmium* $\mu^{Tri}$ is computed as follows:
$$\mu^{Tri} = \mu_{FixN_2} + \mu^{Tri}_{NO_3} + \mu^{Tri}_{NH_4} \tag{2}$$
where $\mu_{FixN_2}$ denotes growth due to dinitrogen fixation, $\mu_{NO_3}^{Tri}$ and $\mu_{NH_4}^{Tri}$ represent growth sustained by $NO_3^-$ and $NH_4^+$ up-
take, respectively. Moreover, a fraction of fixed nitrogen is released back to seawater, mainly as ammonia and dissolved organic
nitrogen, by the simulated *Trichodesmium* compartment. Berthelot et al. (2015a) estimated this fraction to be less than 10%
when considering all diazotrophs. We set up this fraction at 5% of the total amount of fixed nitrogen. For the other nutrients
(i.e. iron and phosphorus), the same fraction is also released.
Dinitrogen fixation is limited by the availability of phosphate, iron and light and is modulated by temperature. Loss processes
are natural mortality, and grazing by zooplankton. Natural mortality is considered to be similar to the other modeled phyto-
plankton species. Grazing on *Trichodesmium* is rarely described, but it is admitted that *Trichodesmium* represents a poor source
of food for zooplankton (O'Neil and Romane, 1992) especially because they contain toxins (Hawser et al., 1992). On the other
hand, many species of copepods have been shown to be able to graze on *Trichodesmium* despite the strong concentrations
of toxins (O'Neil and Romane, 1992). For these reasons we applied two different coefficients for the grazing preference by
mesozooplankton and microzooplankton (Table 1). For microzooplankton, grazing preference is halved to account for *Tri-*
*chodesmium* toxicity, and for mesozooplankton the grazing preference is similar to that of the other phytoplankton species.
The complete set of equations of *Trichodesmium* is detailed in Appendix 1. Table 1 presents the parameters that differ between
Nanophytoplankton and *Trichodesmium*.
This setup reproduces dinitrogen fixation through an explicit representation of the *Trichodesmium* biomass (to be compared
with often used implicit parameterizations (Assmann et al., 2009; Aumont et al., 2015; Dunne et al., 2013; Maier-Reimer
et al., 2005; Zahariev et al., 2008) that links directly environmental parameters to nitrogen fixation without requiring the *Tri-*
*chodesmium* biomass to be simulated).

## 2.2 Experimental setup

Below are summarized the set of experiments that have been performed in this study (Table 2). All climatological simulations
have been run for 20-years from the same restart and only the last 19-years are considered in our diagnostics. We chose the
above described simulation explicitly modelling the *Trichodesmium* to be our reference experiment (hereafter referred to as
«TRI»). In a second experiment called «TRI_NoFeSed», the model setup is identical to the reference experiment, except that
iron input from the sediments is turned off between 156°E and 120°W. In a third experiment «TRI_imp», the explicit dinitrogen
fixation module is replaced by the implicit parameterization described in Aumont et al. (2015) where fixation depends directly
on water temperature, nitrogen, phosphorus and iron concentrations and light (no nitrogen fixers are simulated). Finally, a
fourth experiment «N2_Wo» corresponds to a model setup in which no explicit nor implicit description of dinitrogen fixation
is activated.
Comparison between TRI and TRI_NoFeSed experiments enables to estimate the impact of iron input from island sediments
on the dinitrogen fixation, while the impact of dinitrogen fixation on the biogeochemical conditions in the Pacific Ocean can
be investigated by comparing TRI and N2_Wo. Finally, the TRI and TRI_imp experiments are used to evaluate the added value
of an explicit description of dinitrogen fixation relative to an implicit inexpensive parameterization.

## 2.3   Observational datasets

Several different databases have been used to evaluate the model skills. For nitrate and phosphate, the CSIRO $1/2°$ global Atlas of Regional Seas (CARS, http://www.marine.csiro.au/ dunn/cars2009/) has been used. Iron has been evaluated with the global database from Tagliabue et al. (2012) complemented with the dissolved iron data from the OUTPACE cruise (Guieu et al., under review). This database is a compilation of 13125 dissolved iron observations covering the global ocean and encompassing the period 1978–2008. The global MARine Ecosytem DATa (MAREDAT, https://doi.pangaea.de/10.1594/PANGAEA.793246) database of $N_2$ fixation has been expanded with data from recent cruises performed in the WTSP: MOORSPICE (Berthelot et al., 2017), DIAPALIS (Garcia et al., 2007), NECTALIS (http://www.spc.int/oceanfish/en/ofpsection/ema/biological-research/nectalis), PANDORA (Bonnet et al., 2015), OUTPACE (Bonnet et al., This issue), Mirai (Shiozaki et al., 2014). This database contains 3079 data points at the global ocean scale, of which  1300 are located in our simulated region (Luo et al., 2012). Finally, we have used surface chlorophyll concentrations from the GLOBCOLOUR project (http://ftp.acri.fr) which spans the 1998-2013 time period.

## 3   Results

### 3.1   Model Validation

In this subsection, we aim at validating our reference simulation «TRI» with the data previously presented. In the Pacific, phosphate and nitrate concentrations show maxima in the upwelling regions, i.e. along the western American coast, and in the equatorial upwelling (Fig. 1a,c), and mimima in the subtropical gyres. First, phosphate patterns show modeled values and structures in qualitatively good agreement with observations, despite an underestimation in the areas of high concentrations as within the Costa Rica dome and along the equator. In contrast, the nitrate structure shows larger biases. We observe concentrations higher than 1 $\mu$mol.$L^{-1}$ all along the equator in CARS, while in the model nitrate concentrations are lower than this value west of 170W°. More generally the model tends to underestimate nitrate concentrations.

The regions most favorable for *Trichodesmium* can be defined by temperature within 25-29°C (Breitbarth et al., 2007). The model reproduces relatively well the spatial distribution of this temperature preferendum. This distribution exhibits a significant seasonal variability, mainly as a result of the variability of the 25°C isotherm. The latter moves by  7° latitudinally between summer and winter in the WTSP, and by  15° in the Western Tropical North Pacific (WTNP) (Fig. 1a). This displacement is well reproduced in the TRI simulation (Fig. 1b). By contrast, along the equator the mean position of the 25°C isotherm is shifted eastward in the TRI simulation (120°W) compared to the observations (125°W; Fig. 1a vs 1b), but its seasonal displacements are well reproduced except in the South-Eastern Pacific. Overall, this temporal variability is well reproduced by the model (Fig. 1b), despite this bias. In contrast, nitrate and phosphate seasonal variability remains low (not shown).

Another important feature that needs to be properly reproduced by the model is the iron distribution in the upper ocean. We have sampled the modeled values at the same time and same location as the data. The median value as well as the dispersion of the iron surface concentrations over the tropical Pacific, are displayed for both the data and the model in Figure 2a.

Mann-Whitney test reveals that these two normalized distributions are not significantly different (p-value = 0.26). Figures 2b,c
display respectively the observed iron field and the modeled values. The best sampled area is the central Pacific ocean where
simulated iron concentrations are low (0.1 to 0.3 nmol Fe.L$^{-1}$, Fig. 2c), which is consistent with the observations (Fig. 2b).
The southwest Pacific is characterized by relatively high surface iron concentrations, between 0.4 and 0.8 nmol Fe.L$^{-1}$, both
in the data and in the model. Large scale patterns are thus well represented by the model. Nevertheless, the model tends to
overestimate iron levels in the South Pacific gyre, between 180° and 140°W at about 20°S.
Figure 3 displays a comparison between surface Chlorophyll concentrations from GLOBCOLOUR data (a), and from TRI
(b) and TRI_imp (c) simulations. High chlorophyll concentrations are found in the eastern equatorial Pacific upwelling and
along Peru in both the observations and our two simulations, with mean values higher than 0.3 mg Chl.m$^{-3}$. The equatorial
chlorophyll maximum simulated by the model (Fig. 3b,c) is however too narrow compared to the observations, especially in
the northern hemisphere. Similarly, the model is unable to simulate the elevated chlorophyll levels around the Costa Rica dome
and the localized enhanced chlorophyll off Papua New Guinea. In TRI (Fig. 3b), chlorophyll values in the South West Pacific
region vary between 0.1 and 0.2 mg Chl.m$^{-3}$, with maxima located in the vicinity of the Fiji and Vanuatu islands. These values
are within the range of the data, even if the data tend to be slightly higher (up to 0.3 mg Chl.m$^{-3}$ near the coasts). The spa-
tial structure is well represented, with maxima simulated around the islands. In the subtropical gyres, the simulation predicts
chlorophyll concentrations of 0.05 mg Chl.m$^{-3}$ which are higher than in the observations (< 0.025 mg Chl.m$^{-3}$). In contrast
in TRI_imp (Fig. 3c), chlorophyll values in the South West Pacific and in the North hemisphere are too low in comparison with
the ocean colour data (Fig. 3a).
Part of the surface chlorophyll in Figure 3b is associated with *Trichodesmium*. Figure 3d shows the annual mean spatial distri-
bution of surface *Trichodesmium* chlorophyll in the TRI simulation. This distribution displays two zonal tongues in the tropics,
one in each hemisphere. Maximum values are located in the South West Pacific (around Vanuatu archipelago, New Caledonia,
Fiji, and Papua New Guinea(PNG)) and around Hawaii, where they reach 0.06 mg Chl.m$^{-3}$. In the South Pacific, high chloro-
phyll biomass extends eastward until 130°W. Further east, concentrations drop to below 0.02 mg Chl.m$^{-3}$. It is important to
note that, in the observations, *Trichodesmium* have never been observed eastward of 170°W. This bias in the model could be
explained by the overestimated iron concentrations in the South Pacific Gyre (SPG). In the Northern Hemisphere, between
the coasts of Philippines (120°E) and Hawaii (140°W), *Trichodesmium* chlorophyll concentrations are greater than 0.03 mg
Chl.m-3. In the North East Pacific, *Trichodesmium* chlorophyll is lower, yet significant (<0.03 mg Chl.m$^{-3}$). Otherwise the
equatorial Pacific and South East Pacific oceans are overall poor in *Trichodesmium*.
In Figure 4, the dinitrogen fixation rates predicted by the model in TRI are compared to the observations from the MAREDAT
expanded database. Evaluation of the model behavior remains quite challenging because of the scarcity of the observations.
Some large areas are not properly sampled such as the north west tropical Pacific and the eastern Pacific. In addition, some
areas are sampled only in the surface layer (0-30m), while others have been sampled deeper. This non-homogeneous sam-
pling may alter the distribution of the N$_2$ fixation rates and undermine the comparison with model outputs. To overcome this
sampling bias we compared the observations with N$_2$ fixation rates simulated integrated over two different layers (0-30m and
0-150m). Despite their scarcity, some regional patterns emerge from the observations. Maximum fixation rates (600 to 1600
$\mu$mol N.m$^{-2}$.d$^{-1}$; Fig. 4a) are observed around the south west Pacific islands, in the Solomon Sea, around the Melanesian
archipelagos and around Hawaii, four well known « hotspots » of N$_2$ fixation (Berthelot et al., 2015b, 2017; Bonnet et al.,
2009, 2017; Böttjer et al., 2017). The modeled regional patterns of strong fixation are coherent with the observations (Fig.
4b), showing values in the same range. In the south Pacific, the TRI simulation is able to reproduce the strong east-west in-
creasing gradient of N$_2$ fixation (Shiozaki et al., 2014; Bonnet et al., This issue; Fig 4c,d). In the equatorial Central Pacific,
modeled values of mean fixation are negligible (< 0.5 $\mu$mol N.m$^{-2}$.d$^{-1}$) in contrast to the observations which suggest low but
non-negligible fixation rates (between 1 to 2 $\mu$mol N.m$^{-2}$.d$^{-1}$) (Bonnet et al., 2009; Halm et al., 2012). On the whole mod-
eled domain and for both integration layers, dinitrogen fixation rates are overestimated by 70% in TRI compared to the data.
Some recent studies have shown that the 15N$_2$-tracer addition method (Montoya et al., 2004) used in most studies reported
in the MAREDAT database may underestimate N$_2$ fixation rates due to an incomplete equilibration of the 15N$_2$ tracer in the
incubation bottles. Thus, this overestimation may be an artifact arising from methodological issues (Großkopf et al., 2012;
Mohr et al., 2010). However, some other studies performed in the South Pacific (Bonnet et al., 2016b; Shiozaki et al., 2015)
compared the two methods, and did not find any significant differences.

### 3.2 *Trichodesmium* Primary Production

We evaluated the direct relative contribution of *Trichodesmium* to PP (Fig. 5). The spatial distribution of this contribution
is very similar to the spatial distribution of *Trichodesmium* chlorophyll, with 2 distinct tongues located on each side of the
Equator in the tropical domain. In the Northern hemisphere, the tongue extends from the coast of Philippines (120°E) to
Hawaii (140°W) longitudinally and between 10°N and 25°N latitudinally. The maximum contribution (35%) is reached near
Hawaii while in the rest of the tongue, values are close to 20%. In the Southern Hemisphere, the region of elevated contribution
extends from PNG (140°E) to about the center of the South Pacific subtropical gyre at 130°W, and between 5°S and 25°S
latitudinally. Maximum values are predicted in the vicinity of Vanuatu and Fiji Islands, where they can reach 35%. Part of
this elevated contribution is explained by the very low PP rates simulated in this region for both nanophytoplankton and
diatoms (less than 0.03 mol C.m$^{-3}$.yr$^{-1}$). Furthermore, the island effect seems to represent an important factor for explaining
the spatial distribution of *Trichodesmium* growth rates. Indeed, maximum *Trichodesmium* chlorophyll concentrations and the
largest contribution of *Trichodesmium* to PP are achieved near the islands. Finally, in LNLC regions (red boxes; fig. 1c,d), we
assess that *Trichodesmium* contribute to 15% of total PP, which is in accordance with biogeochemical studies performed in
these areas (Bonnet et al., 2015; Berthelot et al., 2017; Caffin et al., This issue)

### 3.3 Seasonal variability of *Trichodesmium* biomass

*Trichodesmium* biomass (Fig. 6) and simulated dinitrogen fixation rates (Fig. 7) display a seasonal variability that is driven
by the seasonal variability of the environmental conditions (light, temperature, currents, nutrients). The regional maxima of
*Trichodesmium* biomass (exceeding 3 mmol C.m$^{-2}$; integrated over the top 100m of the ocean) are found in both hemispheres
during the summer season (Fig. 6a,b) even if locally, maxima can be attained during other periods of the year than summer. In
the south Pacific, the area of elevated *Trichodesmium* biomass moves by 3° southward from austral winter to austral summer.
Along Australia and in the Coral Sea, *Trichodesmium* biomass exhibits a large seasonal variability with a very low winter
biomass that contrasts with elevated values in summer. A similar important variability, which is shifted by six months, is
simulated in the Northern hemisphere in the Micronesia region and in the Philippine Sea.
Unfortunately, due to the scarcity of $N_2$ fixation data, this seasonal cycle cannot be properly assessed at the scale of the tropical
Pacific Ocean. This is only feasible at the time series station ALOHA located in the North Pacific gyre at 22°45', 158°W,
where seasonal data of dinitrogen fixation are available from 2005 to 2012 (Böttjer et al., 2017). They proved that vertically
integrated dinitrogen fixation rates are statistically significantly (one-way ANOVA, $p<0.01$) lower from November to March
(less than 200 $\mu$mol $N.m^{-2}.d^{-1}$) than from April to October (about 263 ± 147 $\mu$mol $N.m^{-2}.d^{-1}$) as highlighted in Figure
7a (blue dots). In the model (red dots; Fig. 7a), the maximum amplitude of the seasonal cycle appears to be underestimated
relative to the observations (i.e. respectively 170 $\mu$mol $N.m^{-2}.d^{-1}$ and 250 $\mu$mol $N.m^{-2}.d^{-1}$). Dinitrogen fixation peaks one
month earlier in the model than in the data (August for the model and September for the data). Indeed, the simulated dinitrogen
fixation rates are minimum between December and May (averaging 241 ± 27 $\mu$mol $N.m^{-2}.d^{-1}$) and maximum the rest of the
year (averaging 347 ± 52 $\mu$mol $N.m^{-2}.d^{-1}$). These values are comparable to the data even if they are slightly higher.
In order to assess the seasonal cycle of $N_2$ fixation rates in the south Pacific (red box Fig 1c; 160°E-230°E; 25°S-14°S), we have
extracted the available data for each month from our database (blue dots; Fig. 7b), and the corresponding model values in TRI
(red dots; Fig 7b). In July no observations are available and in January, April and August, only one data point is available for the
entire region. The predicted seasonal cycle is broadly consistent with the observations. Minimum dinitrogen fixation rates (239
± 205 $\mu$mol $N.m^{-2}.d^{-1}$ ) occur during austral winter and autumn. Maximum rates are reached in February and March, where
they exceed 600 $\mu$mol $N.m^{-2}.d^{-1}$ in the observations. The increase in dinitrogen fixations rates occurs one month earlier than
in the observations, in December instead of January, and remains two to three fold higher from April to June. It is important
to note here that the sampling spatial and temporal distribution may distort the seasonal cycle. Using the model, it is possible
to evaluate how well the seasonal cycle is captured by the sampling (red dots compared to green dots; Figure 7b). The general
structure of the seasonal cycle remains relatively unaltered. However, the amplitude is significantly impacted since it reaches
1100 $\mu$mol $N.m^{-2}.d^{-1}$ if sampled at the observed stations whereas it is about twice as low at 600 $\mu$mol $N.m^{-2}.d^{-1}$ if all the
model data points are considered. We can conclude that the TRI simulation reproduces well the seasonal cycle of $N_2$ fixation
rates at the Pacific scale, even though more data are needed to improve the evaluation of the model skills.
To further investigate the mechanisms that drive the seasonal variability of *Trichodesmium* in the Pacific, we examined the
factors that control *Trichodesmium* abundance in the TRI simulation (not shown). This decomposition shows that the physical
terms (advection and mixing) are negligible compared to biological terms. In addition, the seasonal cycles of grazing and
mortality are in phase with the production terms but their sign is opposite. In conclusion, this analysis indicates that this
seasonal variability is mainly controlled by the levels of primary production, the others terms of tracer evolution dampe its
amplitude but do not change its shape. Hence we further examine the limitation terms of primary production (Fig. 8) in two
representative regions characterized by elevated levels of $N_2$ fixation rates (red boxes; Fig 1c). A detailed description of these
limitation terms is given in Appendix 1. A limitation term reaching 1 means that growth is not limited whereas a limitation
term equal to zero means that growth ceases.
*Trichodesmium* growth sustained by nitrate and ammonia is very small in LNLC regions due to their very low availability
and is therefore not considered further. Thus, our analysis is restricted to dinitrogen fixation. *Trichodesmium* growth can be
limited by iron and phosphate and is inhibited when reactive nitrogen (nitrate and ammonia) is available. In the WTSP, the
model suggests that iron is the sole nutrient that modulates *Trichodesmium* growth (red curve; Fig. 8a,b). The others limiting
factors of *Trichodesmium* growth are light (green curve; Fig 8a,b) and temperature (purple curve; Fig. 8a,b). The product of
these 3 limiting factors gives the limiting coefficient of dinitrogen fixation (brown curve; Fig 8a,b). The limiting factors vary
according to the season and the hemisphere. In the south (north) Pacific, temperature and light are less limiting during the
austral summer (winter) than during the austral winter (summer). The limiting factor associated to temperature varies from
0.8 to 1, and the light limiting factor varies from 0.15 to 0.3. Unlike light and temperature, iron is less (more) limiting in the
south (north) pacific during winter (summer) than during the austral summer (winter) with values varying between 0.4 and 0.7.
Finally, *Trichodesmium* growth is more limited during the austral winter (summer) in the south (north) Pacific. The seasonal
variability is forced by light and temperature, and iron mitigates its amplitude. Indeed, nutrients and iron inputs brought to the
euphotic zone by the seasonally enhanced vertical mixing are counter-balanced by the related inputs (e.g temperature) of these
water masses.
**4  Discussion**
**4.1  Impact of iron from island sediments**
Monteiro et al. (2011) performed a sensitivity study and found that a fivefold increase in the solubility of aeolian iron improves
the biogeographical distribution of $N_2$ fixation in the southwest Pacific. In the meantime, a recent study has challenged this
view by showing no increase in $N_2$ fixation in response to increased dust deposition (Luo et al., 2014). In any cases, the
sedimentary and hydrothermal sources were not taken into account in those studies, although they are likely significant sources
(Bennett et al., 2008; Johnson et al., 1999; Moore et al., 2004; Tagliabue et al., 2010; Toner et al., 2009). In parallel, Dutkiewicz
et al. (2012) evaluated the sensitivity of the biogeographical distribution of $N_2$ fixation to the aeolian source of iron in a model
which takes into account the iron sediment supplies, and conclude for minor changes in the south west Pacific, while in the
north Pacific the change was larger. Indeed, there are many islands with a marked orography that could deliver significant
amounts of iron to the ocean (Radic et al., 2011) in the southwest Pacific.
To assess the impact of the sediment source of iron on the *Trichodesmium* production, we used the «TRI_NoFeSed» experiment
in which this specific source of iron has been turned off for the islands between 156°E and 120°W (Table 1). In this simulation,
iron and *Trichodesmium* chlorophyll decrease by 58% and 51% respectively (Fig. 9a,b), in the WTSP (red box Fig 1c). Figure
9c displays the iron distribution simulated in TRI_NoFeSed, and shows that the maximums around the islands disappear.
Furthermore, in the south Pacific, iron decreases due to the reduction of the zonal advection of iron downstream of the islands.
The iron flux from the sediments around the islands also affects the spatial structure of *Trichodesmium* chlorophyll (Fig.
9e,f), most noticeably in the south Pacific, with maxima shifted from the south Pacific islands region (e.g. Fiji, New Caledonia,
Vanuatu) in the TRI simulation to the coastal regions near Australia and Papua New Guinea in the TRI_NoFeSed simulation. In
the northern hemisphere, the effects of the sediment flux of iron are less important with a shift of the *Trichodesmium* chlorophyll
maxima towards the Philippine Sea and a localized effect near Hawaii. This sensitivity test demonstrates that *Trichodesmium*
are highly sensitive to the iron distribution in our model and hence that the spatial patterns of *Trichodesmium* chlorophyll in
the south west Pacific are tightly controlled by the release of iron from the coastal sediments of the Pacific islands.

## 4.2  *Trichodesmium* impacts on biogeochemistry

One of the questions we want to address is the quantification of the *Trichodesmium* impact on primary production (PP), at
the Pacific scale with a focus on the WTSP region. In the oligotrophic waters of the south pacific, dinitrogen fixation can
be an important source of bio-available nitrogen in the water column through *Trichodesmium* recycling which can feed other
phytoplankton. To evaluate that impact, we calculated the relative increase of PP between the TRI simulation and the N2_Wo
simulation in which no $N_2$ fixation is considered (Fig 10a). As expected, the spatial structure of the primary production differ-
ences between both simulations matches the $N_2$ fixation spatial distribution in the TRI simulation (two tongues, one in each
hemisphere). In the north Pacific the maximum increase of the primary production due to the $N_2$ fixation is located around
Hawaii, where it exceeds 120%. In the remaining part of the northern hemisphere tongue, primary production increases by
50% to 100%. In the southern hemisphere, values are more homogeneous in the tongue (from 80 to 100%), even though there
is a local maximum around Fiji and Vanuatu (up to 120%). Out of these northern and southern tongues, the increase of PP is
less than 20%. In average on our domain, the increase of PP is 19% and in LNLC regions, it reaches approximately 50%.
From total primary production only, it is not possible to disentangle the increase of primary production directly due to *Tri-*
*chodesmium* themselves and the indirect increase due to the impact of $N_2$ fixation on the other phytoplankton groups (nanophy-
toplankton and diatoms). Indeed, as mentioned in the method section, *Trichodesmium* also releases a fraction of the recently
fixed $N_2$ as bio-available nitrogen (in our model Trichodemium releases ammonia and dissolved organic nitrogen, but only
ammonia is directly bio-available). Figure 10b displays the difference in PP due to diatoms and nanophytoplankton only. The
main large scale patterns constituted of the northern and southern tongues persist, but the intensity of the differences contrasts
with those found when considering total primary production (Fig. 10a). Indeed, the increase of total primary production (Fig
10a) in those two tongues is twice as high as when the direct effect of *Trichodesmium* is excluded. This analysis stresses the
importance of the bio-available nitrogen released by diazotrophs as we attribute about half of the total production increase to
this release. Indeed, recent isotopic studies tracing the passage of diazotroph-derived nitrogen into the planktonic food web
reveal that part of the recently fixed nitrogen is released to the dissolved pool and quickly taken up (24-48%) by surrounding
planktonic communities (Berthelot et al., 2016; Bonnet et al., 2016b, a).
With the simulation TRI_imp, we aim at comparing an implicit $N_2$ fixation formulation to the explicit formulation used in TRI.
Figure 10c displays the relative change of total PP between the TRI and the TRI_imp simulations (see Table 2). The implicit
formulation displays a similar spatial distribution to that of the explicit distribution but it is predicting a lower total primary
production, especially in the the southern Pacific where explicit formulation leads to an increase of about 45% in total PP
compared to the one related to the implicit formulation. On average across our domain, total primary production is about  9%
higher when nitrogen fixation is explicitly modeled relative to an implicit formulation.
This difference becomes even weaker (2%) if only primary production by nanophytoplankton and diatoms is considered, with
noticeable differences restricted to the areas of maximum $N_2$ fixation in the southern hemisphere (around the islands). PP
sustained by the release of bio-available nitrogen is thus similar in the TRI and TRI_imp simulations, but an explicit represen-
tation of $N_2$ fixation allows a better description of $N_2$ fixation patterns. Indeed, the areas of intense $N_2$ fixation rates cannot be
properly simulated in the vicinity of the islands, especially in the southern hemisphere, by the tested implicit parameterization.
We also assessed the carbon export (under the euphotic layer, mmol $C.s^{-2}.d^{-1}$; Supp. Figure 1) and the $N_2$ fixation rate (inte-
grated over top to 150m; Supp. Figure 2: panel a in $\mu$mol $N.m^{-2}.d^{-1}$ and panel b in percentage) difference between TRI and
TRI_imp simulations. We observe a carbon export greater in TRI simulation, the average across the Pacific of this difference is
0.1 mmol $C.m^{-2}.d^{-1}$ or 4 %, and in LNLC regions the increase varies between 6 and 10 %. The $N_2$ fixation rates are greater
in TRI simulation except in the warm pool, in the equatorial upwelling, and in Peru upwelling.

## 394 4.3 Limitations of the present study

In this study, we simulate $N_2$ fixation through the explicit representation of only one type of diazotrophs, the *Trichodesmium*
sp. This choice has been motivated by evidences that it represents one of the main nitrogen fixers in the western tropical Pacific
(Bonnet et al., 2015a; Dupouy et al., 2011; Shiozaki et al., 2014) and by the relatively good knowledge (compared to other
dinitrogen fixers) we have about its physiology (Ramamurthy and Krishnamurthy, 1967; Ohki et al., 1992; Mulholland and
Capone, 2000; Mulholland et al., 2001; Küpper et al., 2008; Rubin et al., 2011; Bergman et al., 2013). However, our model
remains simple and some of the mechanisms that drive the behavior of *Trichodesmium* have not been implemented in our model.
As an example, the ability of *Trichodesmium* to group in colonies and to vertically migrate (Kromkamp and Walsby, 1992;
Villareal and Carpenter, 2003; Bergman et al., 2013) is well documented. The reason of these mechanisms remains unclear,
but several hypotheses have been put forward such as avoiding nitrogenase exposition to di-oxygen (Carpenter, 1972; Gallon,
1992; Paerl et al., 1989), or maximizing light (on the surface) and nutrients (at depth) acquisition (Letelier and Karl, 1998;
Villareal and Carpenter, 1990; White et al., 2006), or even increasing the efficiency of the uptake of atmospheric iron (Rubin
et al., 2011). Our model does not represent those processes, nor does it model the resulting vertical migration of *Trichodesmium*.
Moreover, the release of fixed dinitrogen as reactive nitrogen bioavailable to other phytoplanktonic organisms has been set to
a constant value of 5%. This percentage is known to be highly variable and therefore this value is in the lowest range of the
observations. An increase in this value would increase the PP due to nanophytoplankton and diatoms in the TRI simulation, and
thus decrease the relative contribution of *Trichodesmium* to total PP, which would be closer to the last observations (Berthelot
et al., 2017; Bonnet et al., 2017).
Some studies, mostly based on extrapolated in-situ data, aimed at assessing the potential of $N_2$ fixation at global or regional
scale (Codispoti et al., 2001; Deutsch et al., 2001; Galloway et al., 2004). In the western south tropical Pacific, Bonnet et al
(2017) have estimated total $N_2$ fixation at 15 to 19 Tg $N.yr^{-1}$. For the same region, $N_2$ fixation is predicted to amount to 7
Tg $N.yr^{-1}$ in the TRI simulation. As already mentioned, this rather low predicted estimates might be explained by the sole
representation of *Trichodesmium* as nitrogen fixing organisms, which dominate in the western tropical south Pacific (Dupouy et
al., 2011, Stenegren et al., This issue). It has to be noted that other diazotroph groups such UCYN-B and DDAs are abundant in
the WTSP, representing 10-20% of the overall diazotroph community (Stenegren et al., This issue). Moreover, the contribution
of heterotrophic diazotrophic organism is poorly studied and may account for a significant fraction of $N_2$ fixation (Moisander
et al., 2017). Our model estimation has also been computed from monthly averages and is thus not taking into account the
high-frequency variability that may explain at least some of the very high rates of $N_2$ fixation found in the study by Bonnet et
al. (2017). Our assessment based on a model could thus be seen as a lower limit for $N_2$ fixation in the western tropical Pacific.
Moreover, our model shows also a good qualitative agreement with the studies based on observations that focus on the impact
of $N_2$ fixation in tropical oligotrophic waters (Raimbault and Garcia, 2008; Shiozaki et al., 2013). Indeed, in agreement with
those studies, our reference simulation predicts that diazotrophs support a significant part of total PP (15%) in LNLC regions.

## 5  Conclusion

This study describes the spatial and temporal distribution of *Trichodesmium* at the scale of the tropical Pacific ocean, and
investigates the impact of a major diazotroph species (e.g *Trichodesmium* sp.) on the biogeochemistry of this region. Toward
this end, we performed a first 20-year simulation with the coupled 3D dynamical biogeochemical model ROMS-PISCES in
which we embedded an explicit representation of $N_2$ fixation based on *Trichodesmium* physiology. This simulation was shown
to be able to reproduce the main physical (SST) and biogeochemical (nutrients) conditions of the tropical Pacific ocean. This
includes the spatial distribution of surface chlorophyll and $N_2$ fixation.
The validation of this simulation allows us to confidently assess the *Trichodesmium* distribution. The model predicts that areas
favorable to *Trichodesmium* growth extend from 150°E to 120°W in the south Pacific, and from 120°E to 140°W in the north
Pacific, with local optimal conditions around the islands (i.e., Hawaii, Fiji, Samoa, New Caledonia, Vanuatu). This broadly cor-
responds to the LNLC regions where *Trichodesmium* are predicted to be responsible for 15% of total primary production. The
seasonal variability of the *Trichodesmium* habitat is dominantly controlled by SST and light, while iron availability modulates
the amplitude of the seasonal cycle.
In our study we also assess the role played by iron released from the island sediments, and show that this iron source partly
controls the spatial structure and the abundance of *Trichodesmium* in the western tropical south Pacific. However, this region is
in the center of the south Pacific convergence zone which is the largest convective area of the Southern Hemisphere, with rain-
fall exceeding 6 mm.day$^{-1}$, hence it would be interesting to assess the impact of river iron supply on the diazotrophs activity.
In addition, the Vanuatu archipelago and Tonga are located on the ring fire, hence hydrothermal sources could have a strong
impact on $N_2$ fixation. These two iron sources are not yet implemented in our configuration but may improve simulations of $N_2$
fixation in the south western tropical Pacific region. Finally, our explicit simulation of $N_2$ fixation has proven to be higher by
25% (while still in the lower end of estimations from observations) than the more commonly used implicit parameterization.

# 6  Appendix

*Trichodesmium* preferentially fixes $N_2$ at temperature between 20-34°C (Breitbarth et al., 2007). The temperature effect on the growth rate is modeled using a $4^{th}$ order polynomial function (Ye et al., 2012):

$$L_T^{Tri} = \frac{2,32.10^{-5} \times T^4 - 2,52.10^{-3} \times T^3 + 9,75.10^{-2} \times T^2 - 1,58 \times T + 9,12}{0,25} \tag{3}$$

where $0.25 d^{-1}$ is the maximum observed growth rate (Breitbarth et al., 2007). Hence, at 17°C the growth rate is zero and maximum growth rate is reached at 27°C. The *Trichodesmium* light limitation is similar to nanophytoplankton (Aumont et al. (2015)).

From equation 2, we distinguishe 2 cases for the growth rate due to $N_2$ fixation.

If phosphorus is limiting the equation 2 becomes :

$$\mu_{FixN_2} = \mu_{max}^{Tri}.L_I^{Tri}.L_P^{Tri} - (\mu_{NO_3}^{Tri} + \mu_{NH_4}^{Tri}) \text{ with,} \tag{4a}$$

$$L_P^{Tri} = min\left(1, max\left(0, \frac{(\theta^P - \theta_{min}^P) \times \theta_{max}^P}{(\theta_{max}^P - \theta_{min}^P) \times \theta^P}\right)\right) \tag{4b}$$

if iron is limiting :

$$\mu_{FixN_2} = \mu_{max}^{Tri}.L_I^{Tri}.L_{Fe}^{Tri} - (\mu_{NO_3}^{Tri} + \mu_{NH_4}^{Tri}) \text{ with,} \tag{5a}$$

$$L_{Fe}^{Tri} = min\left(1, max\left(0, \frac{(\theta^{Fe} - \theta_1^{Fe}) \times \theta_{opt}^{Fe}}{(\theta_{opt}^{Fe} - \theta_0^{Fe}) \times \theta^{Fe}}\right)\right) \tag{5b}$$

In equation 5b, $\theta_1^{Fe}$ and $\theta_0^{Fe}$ are computed as follows :

$$\theta_1^{Fe} = \theta_0^{Fe} + \alpha.\mu_{FixN_2} \tag{6a}$$

$$\theta_0^{Fe} = \theta_{min}^{Fe} + m \tag{6b}$$

$$\alpha = \frac{1}{\beta} \tag{6c}$$

$\theta^{Nutrients}$ represents the nutrient quota for Fe and phosphorus (i.e, the ratio between iron and carbon concentrations in *Trichodesmium*, for instance). $\theta_{min}^P$, and $\theta_{opt}^{Fe}$ are constants, whereas $\theta^{Nutrients}$ varies with time. The mimimum between $L_{Fe}^{Tri}$ and $L_P^{Tri}$ defines the limiting nutrient. $L_I^{Tri}$ is the limiting function by temperature and light. m is the difference between the maintenance iron (i.e, the intracellular Fe:C present in the cell at zero growth rate) for a diazotrophic growth and a growth on ammonium (Kustka et al., 2003). $\beta$ is the marginal use efficiency and equals the moles of additional carbon fixed per additional mole of intracellular iron per day (Raven, 1988; Sunda and Huntsman, 1997). The demands for iron in phytoplankton are for photosynthesis, respiration and nitrate/nitrite reduction. Following Flynn et al. (1997), we assume that the rate of synthesis by the cell of new components requiring iron is given by the difference between the iron quota and the sum of the iron required

by these three sources of demand, which we defined as the actual minimum iron quota:
$$\theta_{min}^{Fe} = \frac{0,0016}{55.85}\theta_{Tri}^{Chl} + \frac{1,21.10^{-5} \times 14}{55,85 \times 7,625}L_N^{Tri} \times 1,5 + \frac{1,15.10^{-4} \times 14}{55,85 \times 7,625}L_{NO_3}^{Tri} \tag{7}$$

In this equation, the first right term corresponds to photosynthesis, the second term corresponds to respiration and the third
term estimates nitrate and nitrite reduction. The parameters used in this equation are directly taken from Flynn and Hipkin

479    (1999).

*Acknowledgements.* We thank the ship captains, the scientists as well as funding agencies of all the projects (OUTPACE, BIOSOPE, MOOR-
SPICE, DIAPALIS, NECTALIS, PANDORA, Mirai) that allowed data collection without which we could not validate our model. The authors
thank Institute of Research for Development for supporting all authors. Cyril Dutheil is funded by European project INTEGRE.

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

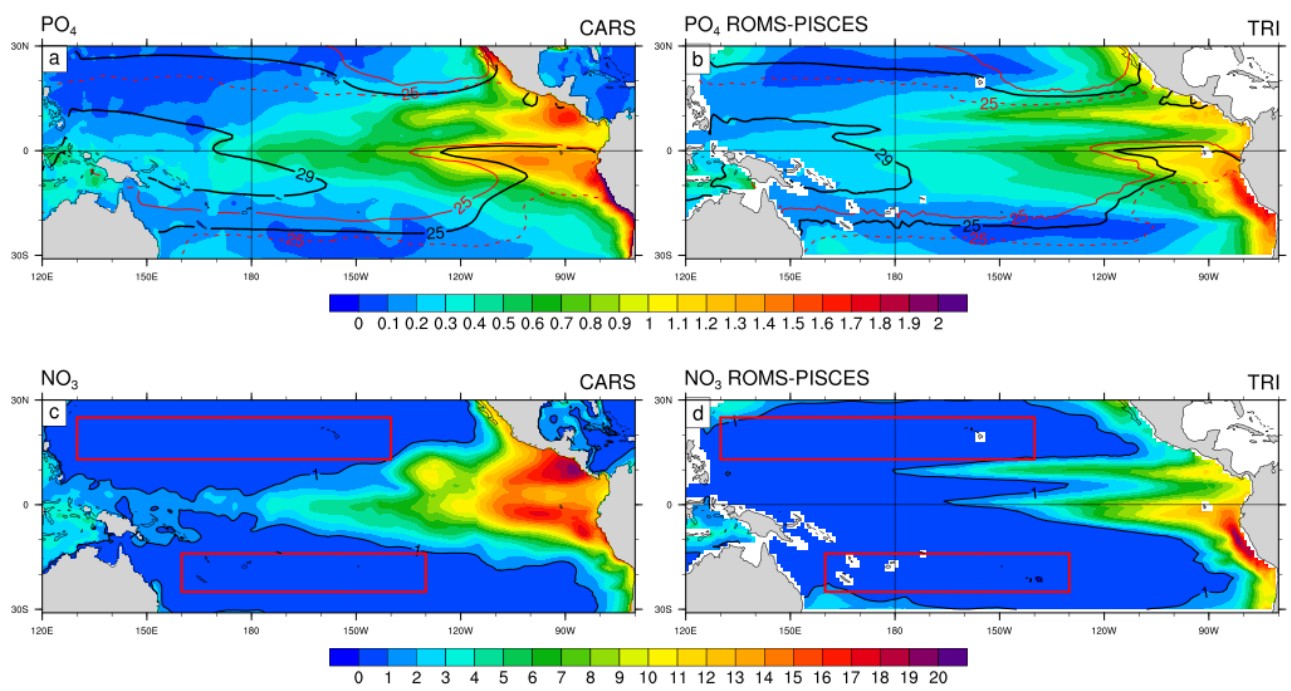

**Figure 1.** Annual mean concentrations in $\mu$mol.L$^{-1}$: a) PO4 data from CARS b) PO4 simulated by the ROMS-PISCES model c) NO3 data from CARS d) NO3 simulated by the ROMS-PISCES model. On panels (a) and (b), the black contours show the annual mean patterns of the temperature preferendum from observations (a) and the model (b). The red contours display the 25°C isolign in austral winter (plain) and in austral summer (dash). On panels (c) and (d) the red boxes represent the LNLC regions (defined as region where [NO$_3^-$] < 1 $\mu$mol.L$^{-1}$ and [Chl] < 0.1 mg Chl.m$^{-3}$ ).

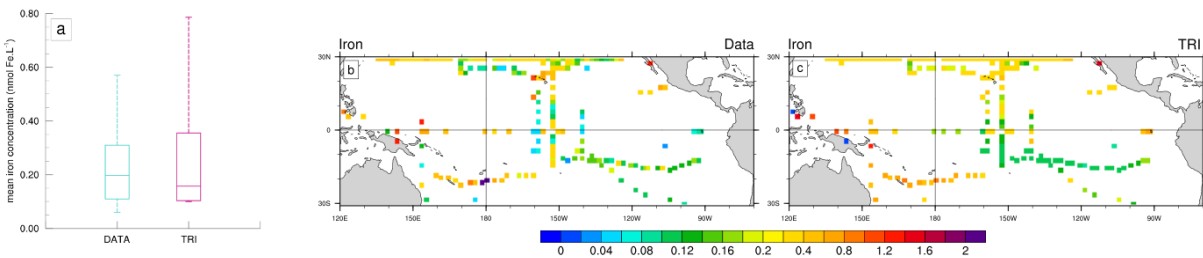

**Figure 2.** Left : Boxplots of the 0-150m averaged Iron (nmol Fe.L$^{-1}$) data (blue) and the equivalent for the model (red) colocalised with the observations in space and time. The coloured box represents the 25-75% quartile of the distribution, the whiskers the 10-90% percentile distribution. The line inside the coloured box is the median.

Right : Iron concentrations (nmol Fe.L$^{-1}$) as observed (b) and as simulated by the model (c). Iron concentrations have been averaged over the top 150m of the ocean. Model values have been sampled at the same location, the same month, and the same depth as the data.

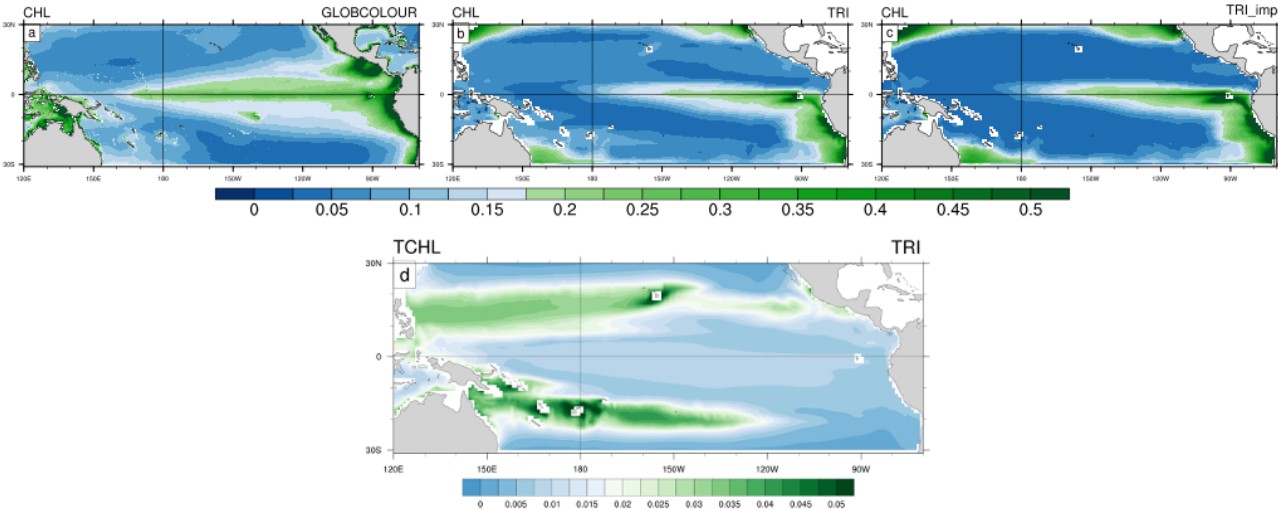

**Figure 3.** Top : Annual mean surface Chlorophyll concentrations (in mg Chl.m$^{-3}$) from (a) GLOBCOLOUR data (b) TRI simulation and (c) TRI_imp simulation.

Bottom: panel (d) shows the the annual mean surface chlorophyll concentrations of *Trichodesmium* in the TRI simulation.

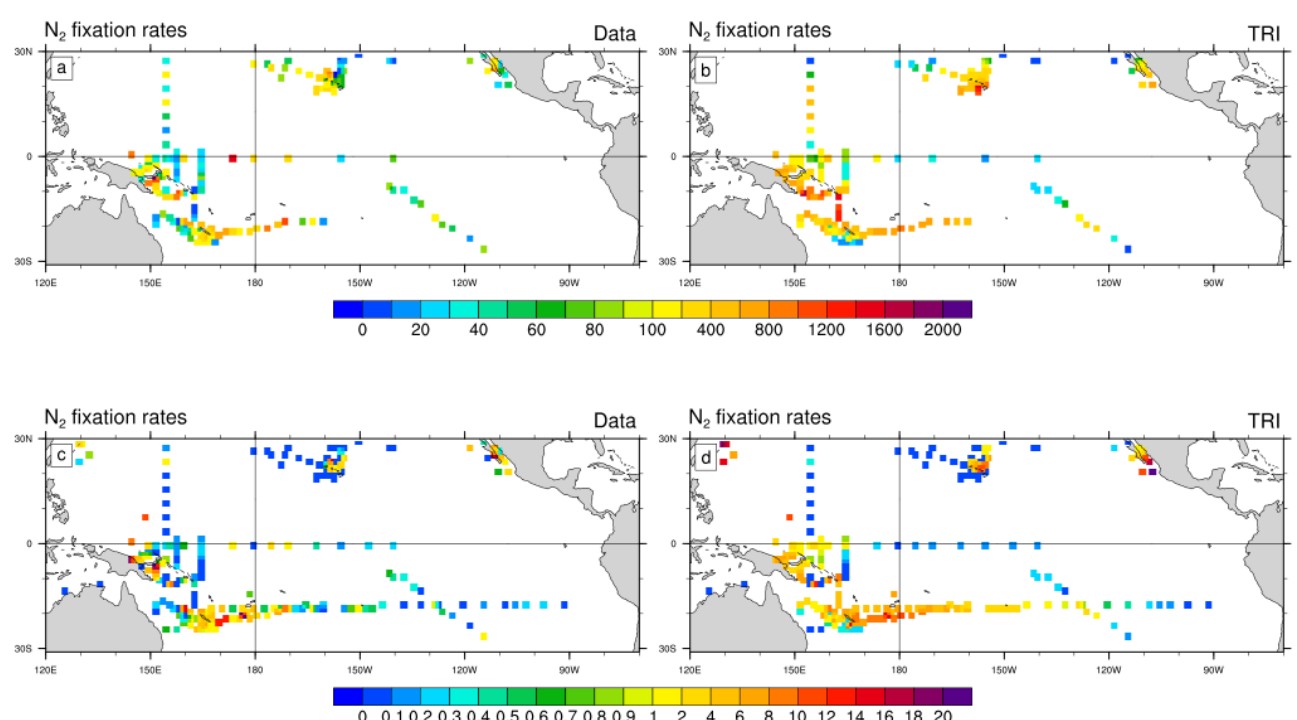

**Figure 4.** N$_2$ fixation rates ($\mu$mol N.m$^{-2}$.d$^{-1}$) as observed (left) and as simulated by TRI simulation (right). In the top panels, N$_2$ fixation rates have been integrated over the top 150m of the ocean. In the bottom panels, the vertical integration has been restricted to the top 30m of the ocean. Model values have been sampled at the same location, the same month (climatological month vs real month), and the same depth as the data.

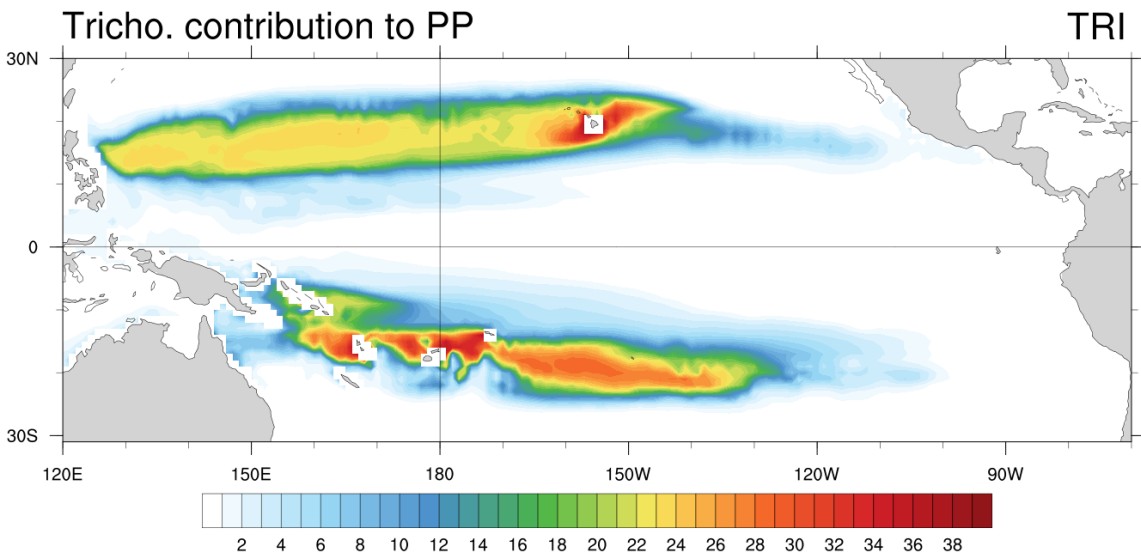

**Figure 5.** Relative contribution (in percentage) of *Trichodesmium* to total primary production.

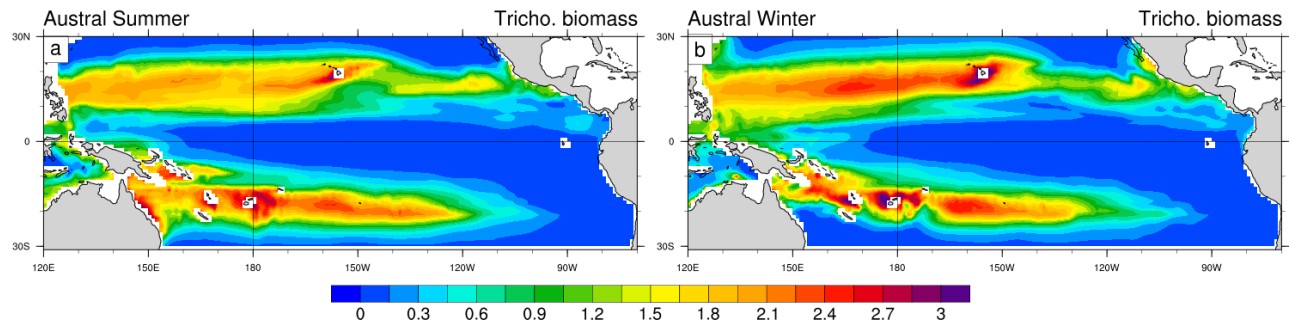

**Figure 6.** *Trichodesmium* biomass (mmol C.m$^{-2}$ ) in (a) austral summer and (b) austral winter, integrated over the top 100m of the ocean.

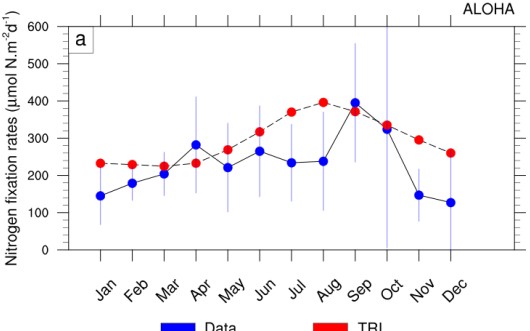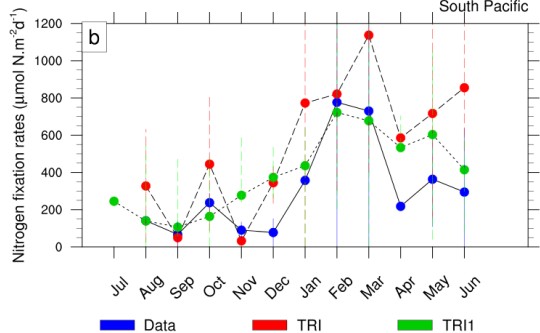

**Figure 7.** a) Depth-integrated (0 to 125m) rates of $N_2$ fixation ($\mu$mol N.m$^{-2}$.d$^{-1}$) at ALOHA for the data (blue) and TRI simulation (red). b) Depth-integrated (from 0 to 150m) rates of $N_2$ fixation ($\mu$mol N.m$^{-2}$.d$^{-1}$) in the south Pacific (red box, Fig. 1c) in the data (blue) and in the TRI simulation (red). The blue curve is the average of all the model points inside the south Pacific zone (red box, Fig. 1c), whereas the green curve corresponds to the average of the model points where data are available.

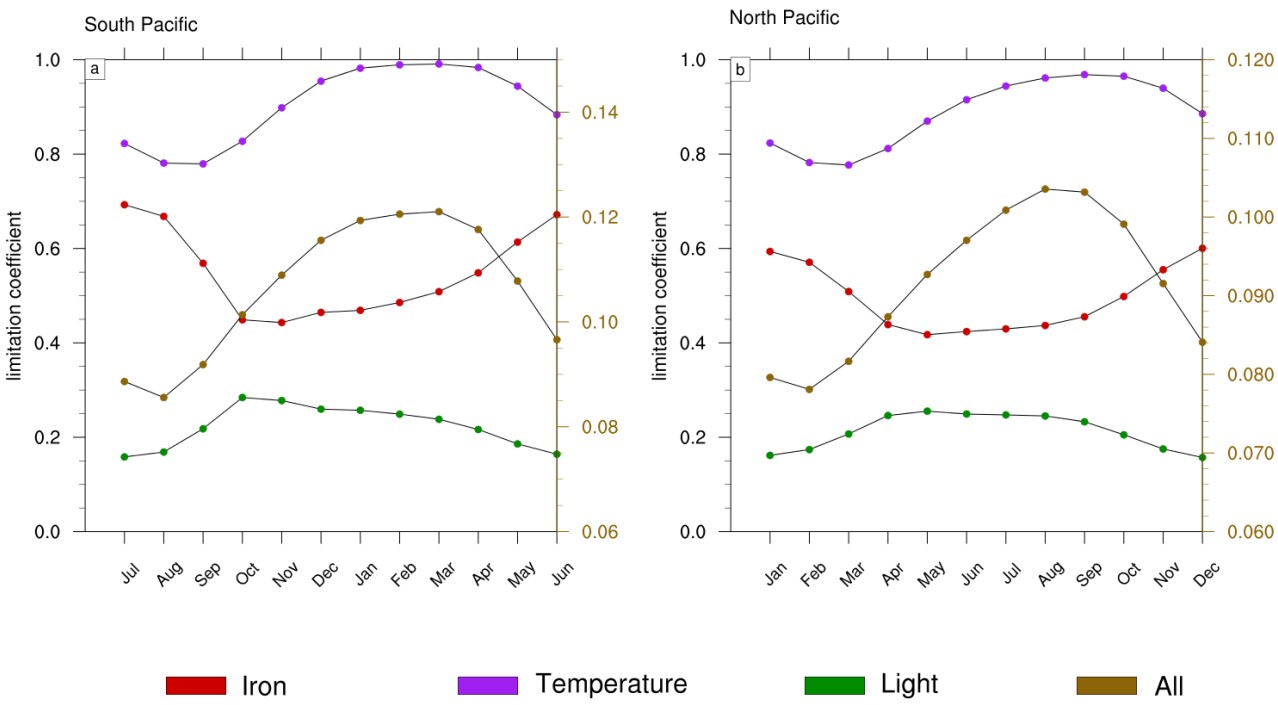

**Figure 8.** Seasonal cycle of the limitation terms of *Trichodesmium* production in a) the South Pacific and b) the North Pacific. The right scale (in brown) represents the total limitation.

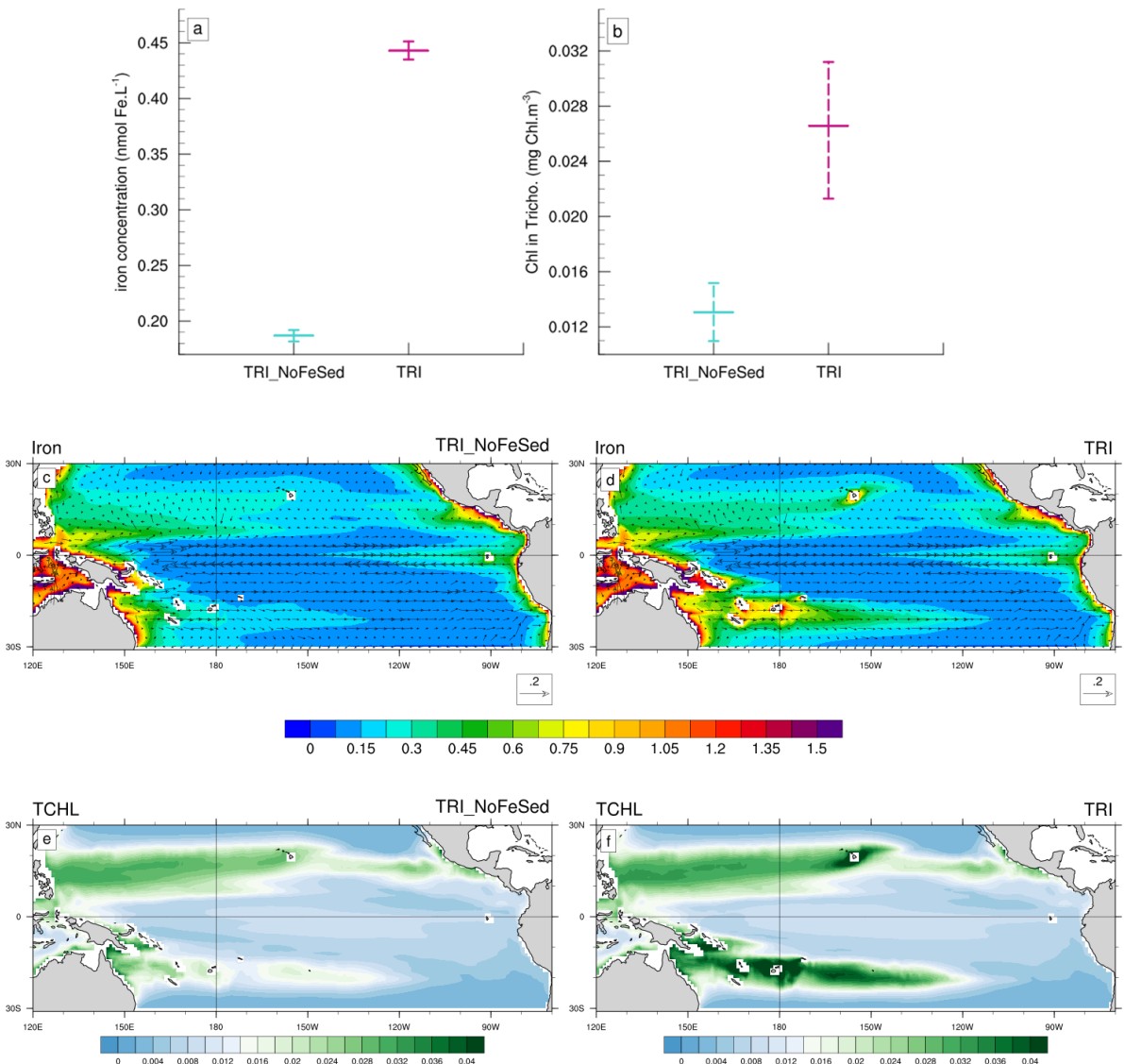

**Figure 9.** Top : Minimum, mean and maximum in the South box (Fig 1c) of (a) the iron concentrations (in nmol Fe.L$^{-1}$), and (b) of the chlorophyll concentrations of *Trichodesmium* (in mg Chl.m$^{-3}$).

Bottom : Annual mean iron concentrations (shading ; in nmol Fe.L$^{-1}$) and current velocities (vectors ; in m.s$^{-1}$) for c) the TRI_NoFeSed simulation and d) the TRI simulation. Annual mean Chlorophyll concentrations of *Trichodesmium* (mg Chl.m$^{-3}$) for e) the TRI_NoFeSed simulation and f) the TRI simulation. The concentrations have been averaged over the top 100m of the ocean. The current velocities are identical on the panels a and b.

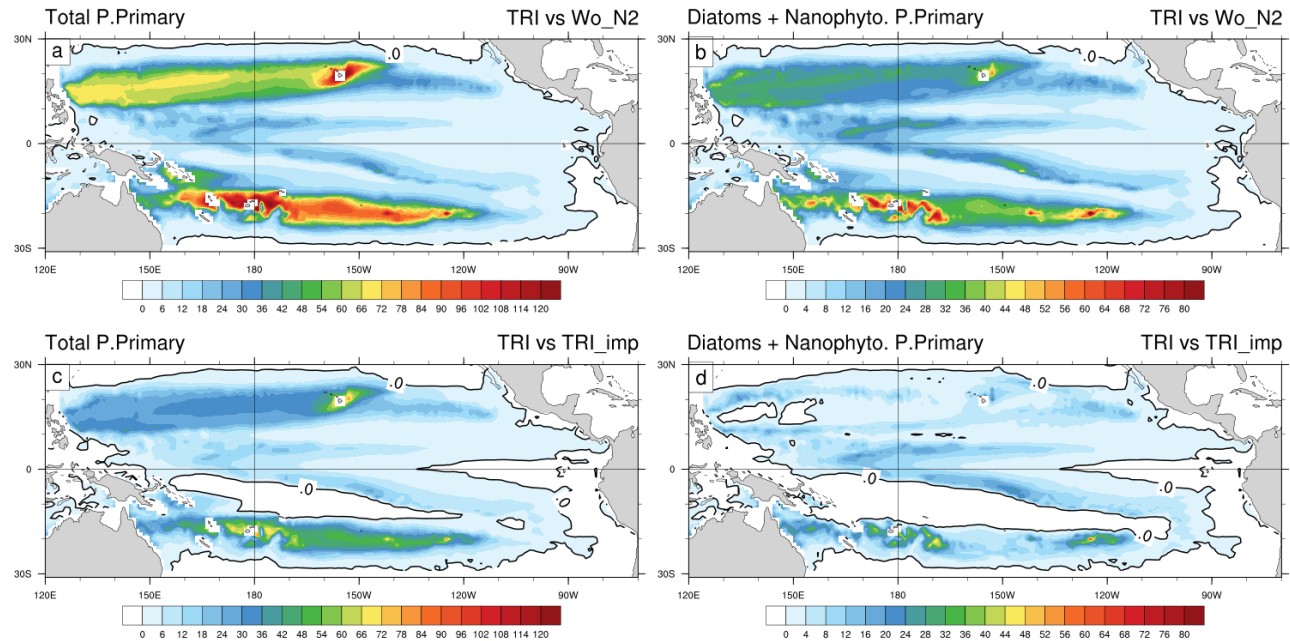

**Figure 10.** Percentage increase of primary production between the TRI simulation and the Wo_N2 simulation (top) and the TRI_imp simulation (bottom); The left panels show total primary production including the contribution of *Trichodesmium* whereas in the right panels, primary production only includes the contribution of diatoms and nanophytoplankton.

**Table 1.** Models parameters for Trichodemium and nanophytoplakton.

| Parameters | Symbol | Unity | Value | Reference |
|---|---|---|---|---|
| Maximum growth rate for Tricho. | $\mu_{max}^{Tri}$ | $d^{-1}$ | 0.25 | Breitbarth et al. (2007) |
| Maximum growth rate for Nano. | $\mu_{max}^{Nano}$ | $d^{-1}$ | 1.0 | |
| Initial slope P-I Tricho. | $\alpha I$ | $(W.m^{-2})^{-1}.d^{-1}$ | 0.072 | Breitbarth et al. (2008) and Hood et al. (2002) |
| Initial slope P-I Nano. | $\alpha I$ | $(W.m^{-2})^{-1}.d^{-1}$ | 2.0 | |
| Microzoo preference for Tricho. | $p_{Tri}^{I}$ | - | 0.5 | |
| Microzoo preference for Nano. | $p_{Nano}^{I}$ | - | 1.0 | |
| Maximum Fe/C in Tricho. | $\theta_{max,Tri}^{Fe}$ | mol Fe.(mol C)$^{-1}$ | $1.10^{-4}$ | Kustka et al. (2003) |
| Maximum Fe/C in Tricho. | $\theta_{max,Nano}^{Fe}$ | mol Fe.(mol C)$^{-1}$ | $4.10^{-5}$ | |
| Maintenance iron | m | mol Fe.(mol C)$^{-1}$ | $1.4.10^{-5}$ | Kustka et al. (2003) |
| Maintenance use efficiency | $\beta$ | mol C.(mol Fe)$^{-1}.day^{-1}$ | $1.4.10^{-4}$ | Kustka et al. (2003) |

**Table 2.** List and description of the different experiments.

| Name configuration | N$_2$ fixation | Iron from sediment |
|:---:|:---:|:---:|
| TRI | explicit | yes |
| TRI_NoFeSed | explicit | no |
| TRI_imp | explicit | yes |
| Wo_N2 | explicit | yes |