# Peer review of "Modeling the *Trichodesmium* sp. related $N_2$ fixation : driving processes and impacts on primary production in the tropical Pacific Ocean"

_Biogeosciences, 2017_

## Referee Comment (RC1) · Anonymous Referee #1 · 8 Mar 2018

General comments The main goal of this study is to model the processes driving the spatial distribution of Trichodesmium N2-fixer in the tropical Pacific Ocean in order to better understand their biogeochemical impact in this oceanic basin. The modelling tool used to achieve this goal is a 3D coupled dynamical-biogeochemical model applied in condition of long-time simulation (20 y). The main innovative point of this study is to develop and validate from field data an explicit formulation of N2 fixation process associated to a Trichodesmium compartment. This formulation is related to the iron intracellular quota in a sophisticated way. In such a configuration the coupled model is able to reproduce, at spatial large scale, the main physical and biogeochemical patterns observed in the dedicated dataset. The assessment of seasonal cycles as that

of N2 fixation rates is less efficient by the model, may be owing to the use of a climatological physical forcing and spatial low resolution of physical model. At the end of this study, a set of sensitivity tests is presented (i) to check the added value arising from an explicit formulation of the N2 fixation process in the coupled model and (ii) to assess the potential roles of iron fluxes from island sediments on the spatial distributions and biomasses of Trichodesmium in the WTSP. On the whole these sensitivity tests are interesting because they enable to increase our knowledge on the biogeochemical roles of Trichodesmium in the tropical Pacific Ocean and can suggest interesting goals for future field cruises. While this paper is of significant scientific interest, it appears uneven in its writing and its quality. Some sections as introduction, discussion and conclusion are clear and well-written, other ones (i.e. methods, results and appendix especially) show many unclear points or they lack of information crucial to a clear understanding. Therefore, I recommend this manuscript for a publication in BG only following major revisions and thorough answers to my requests. Hereafter I give a set of comments, which will somewhat help Authors to improve their manuscript.

Specific comments

1. Title.

The second part of the title does not really make sense. The biogeochemical impact of what on what exactly? Thus I suggest a slight change on this title as for example 'and the biogeochemical impact of N2 fixation on primary productivity in the tropical Pacific Ocean'.

2. Abstract.

- L24-25: I suggest replacing the word 'compartment' by 'parameterization' or 'formulation'.

- L27: Replace the word 'conditions' by 'fields'. Used in this context, the former term is ambiguous. I guess the author would rather mean 'field data/observations'.

[Figure]

- L34: Replace the sentence '. . .the spatial distribution and the abundance of. . .' by 'the spatial distribution of Trichodesmium biomasses in. . .' Your model does not provide abundance (number of cells) of Trichodesmium but rather its biomass.

3. Introduction. - L52-53: redundant reference to Zehr and Bombar (2015).

- L57: missing word (may be 'known'?) in '. . . is consequently better (Bergman et al., 2013. . .'

- L77-79: Author writes about numerical models without indicating references on these models. I would like to see some references in the revised version of the ms.

- L103: '. . . implications for and. . .'

On the whole, the state-of-art on how the N2 fixation process is represented in marine plankton models is lacking in this introduction. Some words on this point are given at the beginning of discussion but I think this piece of text should be rather put and developed also in Introduction. Major point to be addressed.

4. Methods.

- L106: the terms 'primary production model' for PISCES are too restrictive. PISCES is a plankton community model. I suggest replacing 'primary production' by 'biogeochemical'.

L115-127: What are exactly the initial, boundary conditions and forcing of the simulation? This point is not clear to me. Indeed, a climatological forcing strategy seems to be used while it is written in the section 2.2. that the reference simulation is launched over twenty years from 1993 to 2003 suggesting the use of realistic physical (and biogeochemical ?) forcing. No information is given on the types of biogeochemical forcing at the model boundary (e.g. atmospheric deposition of nutrients?). This point needs to be clarified. Major point to be addressed.

- L128-155 on the description of the PISCES model and the formulation of the N2-

fixation process. This section has numerous unclear points and omissions weakening the paper as a whole. This section needs to be reworked and strengthened. Major point to be addressed.

- L137: The reference of Kwiatkowski et al. (subm.) is missing in the bibliography while it is a crucial reference to see the so-called quota version of PISCES model. On the web, this reference cannot be found. Is this paper published now?

- L129-141. The description of the model is too succinct. Is there a term of natural mortality on Trichodesmium? What about the trophic interactions between zooplankton and Trichodesmium? Is zooplankton able to graze Trichodesmium or not? From the Table 1 and the parameters inside I can suppose yes but it is not clearly stated and discussed in this section. The two types of zooplankton seem to be able to graze Trichodesmium. No argument is given for this choice. I would like to see a text around this point in the revised ms. Major point to be addressed.

- L142-144: please be careful on the use of the term 'growth rate'. Here, I suppose it is the photosynthesis growth rate and it is not the net growth rate (gains minus losses) of Trichodesmium. Losses are for example grazing or exudation. Please clarify this point.

- L148-150: What is the form of nitrogen released by Trichodesmium? Is it nitrate and/or ammonium or dissolved organic nitrogen? No information is given on this feature while it is crucial to understand how Trichodesmium can be a source of nitrogen for the plankton community. Moreover, no further information is given on the ability of Trichodesmium to release other element like phosphorus, iron while they are also expected to be constitutive of this planktonic genus. Why are the explicit state variables of the Trichodesmium compartment in fact? To clarify all that points I suggest to add and comment, in this section, a new figure of schematic diagram showing state variables and processes of the Trichodesmium compartment. Major point to be addressed.

In this section, no reference to table 1 is indicated while it is the table of some parameters of the biogeochemical model. More problematic is that all the choices of parameter

values are nowhere discussed in the paper. No references are given in the Table 1. On what criteria have been chosen the values of all these parameters? Is it an arbitrary choice as the exuded fraction of nitrogen by Trichodesmium or from literature? There is an urgent need to justify the values of each parameter presented. I have also some other remarks on the table 1 (see section on table comments hereafter). As another example we don't know why the preference of microzooplankton for Trichodesmium is higher than that of mesozooplankton. I would like to see in the revised version a detailed section on this point. Major point to be addressed.

Furthermore in the section 2.2 (L159-170), crucial information is lacking about the sediment iron flux while it is a key point debated in the present study. What is the value of this flux and its origin (i.e. literature)? Is it a homogenous spatial flux on the whole model grid or only around some islands? One can imagine for example a decreasing flux from coastal to offshore areas. Neither detailed information nor references to previous works are given. Major point to be addressed.

Section 2.3. An important question arises after reading this part of paper. Have the data of OUTPACE cruise been really used to validate the coupled model? This point is not clear to me because the OUTPACE cruise has been carried out in 2015 year while the reference simulation ends 2013 (L160). Several captions of the figures indicate that 'Model values have been sampled at the same location, the same month and the same depth as data'. So I can't understand how it is possible to use the OUTPACE dataset in this paper to validate model output as suggested l175-176 for iron. Major point to be addressed.

5. Results.

L193. Phosphate patterns. 'in qualitatively good agreement'. Assertion to be moderated. The model strongly underestimates the areas of high concentrations as within the Costa Rica dome and along the equator.

L197. 'Regions most favourable for Trichodesmium can be defined by temperature

within 26-29°C'. What is the criterion behind this statement? Is it from literature or from a model results (highest biomasses of Trichodesmium)? Author refers these two temperature limits to preferendums in the caption of Fig. 1. How are defined these preferendums?

L201-202. I don't agree with the sentence on the good reproduction of seasonal variability of SST along the equator by the model. The 26°C isotherm migrates by 15° eastward in the model from summer to winter but this migration is not observed in the SST field.

L205. Please indicate the type of statistical test (and probability) used to prove the result of 'no statistical differences'.

L214. 'with mean values higher than 0.3 mgChl m-3'.

L221. 'Those localized chlorophyll ... effect'. This sentence should be in discussion.

L224-225. 'TRI simulation thus appears. . .' This sentence should be in discussion.

L232. 'SPG'. Acronym not defined.

L236-253. There is neither clear explanation nor associated analysis why the numerical N2 fixation rates are compared with data over two different integration layers (Fig. 4).

L244. Same as my previous remark on the use of OUTPACE iron data in the validation step of model (section 2.3.). How OUTPACE N2 fixation data '(Bonnet et al., this issue)' can be used to validate the model as the simulation ends 2013?

L247. 'In general, ... compared to data'. This sentence is vague and then confusing. Is an overestimation on the whole modelled domain, or only in one sector especially? Is this statement applicable for the rates depth-integrated 0-150m or 0-30m, or both?

L256. The Figure 6 should be numbered 5 instead 6 because it follows the description of Figure 4.

L261. 'PNG'. Acronym not defined.

L272. Why does Author indicate the term 'not shown' for the simulated N2 fixation rates and Trichodesmium biomass as they are presented on Figures 5 and 7, respectively?

L275. 'Figure (6 a,b)' instead Figure (7a,c).

L284. Same remark as L205.

L286. 'Figure 7a' instead 'Figure 5a'.

L303. Please replace 'in the sampling' by 'in the field observations'.

L306-308. What about the other factors (as grazing or natural mortality if existing)? What is the type of analysis exactly, leading to the conclusion 'the seasonal variability is mainly controlled by primary production'? Please replace the term 'by primary production' by 'by the levels of primary production'.

L312-316. Would it be possible to see (in a new table for example) a synthesis of the modelled values of Trichodesmium growth rates of and a comparison of them with those observed in the field if existing or in lab experiments.

L324-326. 'Indeed, ... temperature'. This sentence should be placed in discussion. Furthermore, this sentence is highly debatable. Please be cautious with the concept of 'ocean dynamics mainly 1D'! Is this feature really achieved anywhere in the ocean? Can the Author firmly prove this assertion in the simulations presented in this study?

6. Discussion.

L369. I would like to see a clear definition of the term 'bio-available nitrogen'. Is it dissolved inorganic forms of nitrogen and/or organic forms also?

L393. 'to that of' instead of 'than'.

L400-403. What is the actual reason for a better modelling of N2 fixation by using an explicit representation of this process in the model? At the place of the manuscript one

can expect a deeper analysis of the results. A thorough comparison of the two types of formulation could be lead to explain clearly why using the explicit formulation is an improvement. Is it due, for example, to the inclusion of Fe internal quota in the formulation of Trichodesmium photosynthesis growth rate? Major point to be addressed.

7. Appendix.

In this section all the equations presented should be numbered for clarity. L L477. I suggest to write 'phosphorus or iron' rather than 'phosphorus and iron'.

L479-480. On which basis (literature, experimental works?) the equations of phosphorus and iron limitations have been stated? Are they new formulations? What physiological processes drive the choice of this formulation? I would like to see information on that point in the revised ms.

L480. 'Nutrient quota for Fe and phosphorus' rather than 'Nutrient quota for Fe and PO4'.

L492. Equation of Trichodesmium growth rate if iron limiting. Same remark as for L479-480. Why the N2 fixation growth rate in case of iron limitation is modelled in this way? It is very difficult to evaluate this formulation without explanation! Furthermore, please be careful in using the term ïĄą that can be confused with the term ïĄąl (initial slope of P. vs. I.). It is not clear to me if the value of $\mu$TRIMAX is of 0.25 d-1? If yes, please clearly indicate in the table of parameters (Table 1) its value and at l473. The term LI is used while undefined.

L518. While the limiting function by temperature is defined previously (L471, LT), the limiting function by light is not presented and it deserves to show it. What is the exact form of the term LI defined by the author?

L521. Please be cautious in using the terms of 'new and regenerated production' in this context. The growth rates of Trichodesmium based on nitrate and ammonium are not strictly speaking new and regenerated productions, respectively. Please reconsider

this sentence and formulate your idea with accuracy.

8. Bibliography.

Please check carefully this section. Many typos and different formats.

9. Tables.

Table 1. Major points to be addressed. Why are only presented the parameters of Trichodesmium and nanophytoplankton? Are the parameters of other living biomass compartments remained unchanged (what is the reference in which the unchanged parameters can be found?)? If yes, why those of nanophytoplankton only have been changed? I would like to see in the revised ms explanations on this point. The column 'Name in the code' is useless (technical details) but adding a column with references for each parameter is essential. The important parameter $\mu$TRIMAX is missing in the table. Please check carefully the units of each parameter. According to the definition of ïĄć given in Appendix (L501-502), its unit cannot be in d-1. Replace the term 'excretion' by 'exudation' for parameters 'rTri' and 'rI'.

10. Figures captions.

Fig.1. Typo: 'preferendum'.

Fig. 2. Typo '0-150m'.

Fig. 5b. 'The green curve is the average of the seasonal cycle…' This sentence is not clear to me. How is this average built exactly? This is no more clearly explained in the corresponding section (l300-301).

---

## Referee Comment (RC2) · A. Oschlies (Referee) · 25 Mar 2018

The manuscript describes model simulations without and with two different parameterisations of nitrogen fixation in the tropical Pacific. Results are compared against observations in the ocean's surface layer, and the degree of realism of the two parameterisations employed is discussed. Inferences are made about the role of diazotrophic nitrogen fixation compared to primary production by ordinary phytoplankton.

Overall, the topic is scientifically very interesting and I found the title and also the abstract very promising, but was then disappointed by the material presented in the manuscript (and the often poor way it was presented) for reasons I will explain be-

low. I am afraid I cannot recommend publication of the manuscript in its present form and think that a very major rewrite and additional and thorough analysis is required. This is beyond what I would normally consider as major revision (and would therefore recommend reject and resubmission). As the issue is tricky with special issues, and because the scientific topic is really interesting and it would be a missed opportunity of not analysing this very carefully, I'm still OK with recommending major submission, but want to stress that 'major' should be taken very seriously.

**1   It is impossible to fully understand what has been done**

The explicit description of N2 fixation by Trichodesmium is provided in the Appendix. I tried hard to understand it, but admit that I failed. There may be typos or unexplained terms (e.g., what is $L_{Tri}^{N}$ in line 492? Why are there two different definitions of $L_{Tri}^{Fe}$, lines 479 and 499?). It does not help, that the notation in table 1 seems to be different from the one in the appendix. There are also steps that are not explained or justified. For example line 483 - why is this procedure applied to Fe but not to P? This makes it impossible to understand what has been done and why. There are other models of diazotrophs out in the literature. How does your model relate to these? Why have you developed a new one (is it new?)? To be useful to the scientific community, this has to be presented in much more detail and put into relation to the existing literature.

The authors claim that implicit parameterizations of N2 fixation are often used in bio-geochemical models (line 32, line 154, in the final sentence of the manuscript they even say 'more commonly'), but do not provide a single reference to support this claim. I think this strong statement that is used and certainly requires references and also a detailed description of this implicit parameterisation in order to allow the reader to understand some of the results (see below), and possibly repeat what has been done here.

The set-up of the physical model is unclear as well. line 111 states that it is based on a nested version. Is there a nested version used here? If so, what is the parent and what the child model? Then, in line 116 ff open boundary conditions are introduced. Do these replace the nesting? What does the sentence in line 118 mean "The use of similar ROMS configurations. . .is validated. . ."?

The configuration of the biogeochemical model is not well described. E.g., line 134: a modified version, which differs in the use of a full quota formation. How is it modified? How does it differ? 'variable' Redfield ratios. The Redfield ratio is always constant and always the same (i.e. the one that Redfield used). Replace by variable C:N:P (:Si : Fe:. . .?) ratios. Is the effect of N2 fixation (and denitrification) on alkalinity included in the model? This would be another biogeochemical impact of N2 fixation that should be reported.

In addition to an improved description of N2 fixation, there should also be a description of the growth of diazotrophs as well as their loss terms (grazing, mortality,. . .) and the fate of the fixed N (loss to DOM? Lifetime?)

line 163. Explain why 156E was chosen as western boundary of the test regions without sedimentary iron input? Doesn't this ensure that there is always iron being supplied from the western boundary of the Pacific Ocean?

**2 The presentation of the results is often poor and not as convincing as is could and should be**

Part of this a language problem. Despite the impressive author list, no careful proofreading seems to have taken place before submission. There are many typos, incorrect words, wrong grammar and incomplete sentences. This can (and should) be improved.

Some explanations are very vague and, at closer inspection, are not that convincing.
For example, line 231/232: The bias 'beyond' (presumably 'eastward of'?) 170W is explained by a bias in iron concentrations, which, however occurs mostly west of 150W according to Fig.2.

Fig. 4 Why show the vertical integral and the vertical average in separate panels? The information looks very similar. Explain what differences the reader should see and understand.

One motivation mentioned in the introduction was the comparison of biogeochemical controls and impacts between implicit and explicit representation of N2 fixation. The only comparisons shown are for surface chlorophyll (quite different) (Is the implicit diazotrophic biomass of the implicit representation included here?) and primary production (very similar). Both variables are biogeochemically among the less relevant ones. Showing a comparison for N2 fixation rates, nutrient concentrations, export production, pCO2 and possibly oxygen would be much closer to the original goal of the paper. In my view, such a comparison is essential.

Fig. 2. Why does the run N2_imp have more chlorophyll along the eastern boundary and along the equator than run TRI? This is interesting and might point to some feedbacks in the system.

The comparison among modeled and measured iron concentrations in Fig.2 is very difficult to see. Try different figure types (larger blobs, overly observed 'blobs' on modeled map,...) Same for Fig.4

Fig. 9. Are currents on panels c and d different?

**3   minor points:**

line 326 'cools temperature' is wrong. either lowers temperature or cools the water. line 349. What is meant by high islands?

---

## Author Comment (AC1) · 23 Apr 2018

**General comments :** The main goal of this study is to model the processes driving the spatial distribution of Trichodesmium N2-fixer in the tropical Pacific Ocean in order to better understand their biogeochemical impact in this oceanic basin. The modelling tool used to achieve this goal is a 3D coupled dynamical-biogeochemical model applied in condition of long-time simulation (20 y). The main innovative point of this study is to develop and validate from field data an explicit formulation of N2 fixation process associated to a Trichodesmium compartment. This formulation is related to the iron intracellular quota in a sophisticated way. In such a configuration the coupled model is able to reproduce, at spatial large scale, the main physical and biogeochemical patterns observed in the dedicated dataset. The assessment of seasonal cycles as that of N2 fixation rates is less efficient by the model, may be owing to the use of a climatological physical forcing and spatial low resolution of physical model. At the end of this study, a set of sensitivity tests is presented (i) to check the added value arising from an explicit formulation of the N2 fixation process in the coupled model and (ii) to assess the potential roles of iron fluxes from island sediments on the spatial distributions and biomasses of Trichodesmium in the WTSP. On the whole these sensitivity tests are interesting because they enable to increase our knowledge on the biogeochemical roles of Trichodesmium in the tropical Pacific Ocean and can suggest interesting goals for future field cruises. While this paper is of significant scientific interest, it appears uneven in its writing and its quality. Some sections as introduction, discussion and conclusion are clear and well-written, other ones (i.e. methods, results and appendix especially) show many unclear points or they lack of information crucial to a clear understanding. Therefore, I recommend this manuscript for a publication in BG only following major revisions and thorough answers to my requests. Hereafter I give a set of comments, which will somewhat help Authors to improve their manuscript.

**Legend for the review :**

In yellow the major comments

In blue our answers

**Response to general comments:**

To answer to your comments we have completely rewritten the method and appendix section. The result section has also been reworked so that the speech is clearer and more precise. We have also strengthened the introduction by more accurately detailing the state of the art of nitrogen fixation in biogeochemical models.

**Specific comments :**

**1. Title**

The second part of the title does not really make sense. The biogeochemical impact of what on what exactly? Thus I suggest a slight change on this title as for example 'and the biogeochemical impact of N2 fixation on primary productivity in the tropical Pacific Ocean'.

Indeed, we decided to change the title to « Modeling the Trichodesmium sp. related $N_2$ fixation : driving processes and impacts on primary production in the tropical Pacific Ocean. »

**2. Abstract**

- L24-25: I suggest replacing the word 'compartment' by 'parameterization' or 'formulation'.

We replaced the word 'compartment' by 'formulation' (L24)

- L27: Replace the word 'conditions' by 'fields'. Used in this context, the former term is ambiguous. I guess the author would rather mean 'field data/observations'.

We replaced the word 'conditions' by 'fields' (L26)

- L34: Replace the sentence '. . .the spatial distribution and the abundance of. . .' by 'the spatial distribution of Trichodesmium biomasses in. . .' Your model does not provide abundance (number of cells) of Trichodesmium but rather its biomass.

As suggested, we removed the reference to « abundance » and we replaced '… spatial distribution and abundance …' by '...spatial distribution of *Trichodesmium* biomasses...' (L33)

**3. Introduction**

- L52-53: redundant reference to Zehr and Bombar (2015).

We removed the second occurrence of the reference. (L51-52)

- L57: missing word (may be 'known'?) in '. . . is consequently better (Bergman et al., 2013. . .'

Indeed, the word 'known' was missing and has been added. (L56)

- L77-79: Author writes about numerical models without indicating references on these models. I

would like to see some references in the revised version of the ms.

This section has been deeply modified and includes now several references concerning both, the implicit (Bisset et al., 1999; Maier-Reimer and Kriest, 2005; Assmann et al., 2010; Aumont et al., 2015) and explicit parameterization (Moore et al., 2004; Dunne et al., 2013) of the N2 fixation (L77-91).

- L103: '. . . implications for and. . .'

We followed the reviewer's suggestion (L117)

On the whole, the state-of-art on how the N2 fixation process is represented in marine plankton models is lacking in this introduction. Some words on this point are given at the beginning of discussion but I think this piece of text should be rather put and developed also in Introduction. Major point to be addressed.

Introduction has been modified and we noticeably chose to follow the reviewer's recommendation and replaced the two sentences referring to numerical models (L119 to 121 in the submitted manuscript) by the following paragraph:

« Numerical models have also been used as they allow to overcome the scarcity of observations that may limit the implementation of the two previous approaches (Aumont et al., 2015; Bissett et al., 1999; Dutkiewicz et al., 2012; Keith Moore et al., 2006; Krishnamurthy et al., 2009; Monteiro et al., 2011; Moore et al., 2013; Tagliabue et al., 2008). They can notably be used to investigate the spatial and temporal variability of dinitrogen fixation and to study its controlling environmental factors. In these models, dinitrogen fixation has been implemented in various ways. Some models use implicit parameterizations (Bisset et al., 1999; Maier-Reimer and Kriest, 2005; Assmann et al., 2010; Aumont et al., 2015) to derive dinitrogen fixation from environmental conditions (mainly nitrate, phosphate and iron concentrations, temperature and light) without explicitly simulating any nitrogen fixing organisms. Alternatively, other models rely on explicit descriptions of diazotrophs (Moore et al., 2004; Dunne et al., 2013) from knowledge acquired on *Trichodesmium* sp during laboratory experiments  sp. (Fennel et al., 2001; Hood et al., 2001; Moore et al., 2001). Noticeably, several modeling studies have been focused on the role of iron in controlling the distribution of diazotrophs and dinitrogen fixation (Keith Moore et al., 2006; Krishnamurthy et al., 2009; Moore et al., 2004; Tagliabue et al., 2008). Indeed, a realistic representation of marine iron concentrations has been stressed as a key factor to adequately simulate the habitat of diazotrophs (Monteiro et al., 2011; Dutkiewicz et al., 2012). » (L77-91).

**4. Methods**

- L106: the terms 'primary production model' for PISCES are too restrictive. PISCES is a plankton community model. I suggest replacing 'primary production' by 'biogeochemical'.

We followed the reviewer's suggestion (L120).

-L115-127: What are exactly the initial, boundary conditions and forcing of the simulation? This point is not clear to me. Indeed, a climatological forcing strategy seems to be used while it is written in the section 2.2. that the reference simulation is launched over twenty years from 1993 to 2003 suggesting the use of realistic physical (and biogeochemical ?) forcing. No information is given on the types of biogeochemical forcing at the model boundary (e.g. atmospheric deposition of nutrients?). This point needs to be clarified. Major point to be addressed.

Indeed, the wording used in this section was confusing and some informations about our simulations were missing. Thus, we splitted the « Coupled dynamical (ROMS)-primary production (PISCES) model »  section in a « ROMS » and « PISCES » sections. The latter will be detailed in a following point of this review while the new « ROMS » section is answering to your present comments.

**« 2.1.1 ROMS**

In this study, we used a coupled dynamical-biogeochemical framework based on the regional ocean dynamical model ROMS (Regional Oceanic Modeling System, (Shchepetkin and McWilliams, 2005)) and the state of the art biogeochemical model PISCES (Pelagic Interactions Scheme for Carbon and Ecosystem Studies). The ocean model configuration is based on the ROMS-AGRIF (Penven et al., 2006) informatic code and covers the tropical Pacific region [33°S-33°N; 110°E-90°W]. It has 41 terrain-following vertical levels with 2-5 m vertical resolution in the top 50 m of the water column, then 10-20 m resolution in the thermocline and 200-1000 m resolution in the deep ocean. The horizontal resolution is 1°. The turbulent vertical mixing parameterization is based on the non-local K profile parameterization (KPP) of (Large et al., 1994). Open boundaries conditions are treated using a mixed active/passive scheme (Marchesiello et al., 2001). This scheme is used to force our regional configuration with monthly climatological large-scale boundary conditions from a ½° ORCA global ocean simulation (details available in Kessler and Gourdeau (2007)), while allowing anomalies to radiate out of the domain. The use of similar ROMS configurations (e.g vertical resolution, mixed active/passive scheme, turbulent vertical mixing parameterization) in the WTSP is largely validated through studies demonstrating skills in simulating both the surface (Jullien et al., 2012, 2014; Marchesiello et al., 2010) and subsurface ocean circulation (Couvelard et al., 2008).

To compute the momentum and fresh water/heat fluxes, we also used a climatological forcing strategy. Indeed, documenting the inter-annual to decadal variability is beyond the scopes of our study, which justifies using climatological forcing fields. A monthly climatology of the momentum forcing is computed from the 1993-2013 period of the ERS1-2 scatterometer stress (http://cersat.ifremer.fr/oceanography-from-space/our-domains-of-research/air-sea-interaction/ers-ami-wind). Indeed, ERS derived forcing has been shown to produce adequate simulations of the Pacific Ocean dynamics (e.g, Cravatte et al., (2007)). A monthly climatology at 1/2° resolution computed from the Comprehensive Ocean–Atmosphere Data Set (COADS; Da Silva et al. 1994) is used for heat and fresh water forcing.  In our set-up, ROMS also forces on line a biogeochemical model using a WENO5 advection scheme (i.e. five order weighted essentially non-oscillatory scheme; Shchepetkin and McWilliams, 1998). After a one year spin-up we stored 1-day averaged outputs for analysis. »

- L128-155 on the description of the PISCES model and the formulation of the N2- fixation process. This section has numerous unclear points and omissions weakening the paper as a whole. This section needs to be reworked and strengthened. Major point to be addressed.

Indeed, this comment is shared by the second reviewer. Therefore we decided to add specific sections, much more detailed, on the PISCES model :

 **« 2.1.2 PISCES**

In this study, we used a quota version of the standard PISCES model (Aumont and Bopp, 2006a; Aumont et al., 2015), which simulates the marine biological productivity and the biogeochemical cycles of carbon and the main nutrients (P, N, Si, Fe). This modified model, called PISCES-QUOTA, is extensively described in Kwiatkowski et al. (2018, in press). Our version is essentially identical to Kwiatkowski's version that included an additional picophytoplankton group, except that this latter group has been removed here and replaced by a *Trichodesmium* compartment. Here we only highlight the main characteristics of the model and the specifics of our model version. Our version of PISCES-QUOTA has then 39 prognostic compartments. As in the standard PISCES version, phytoplankton growth is limited by the availability in five nutrients: nitrate and ammonium, phosphate, silicate and iron. Five living compartments are represented: Three phytoplankton groups corresponding to nanophytoplankton, diatoms, and *Trichodesmium* and two zooplankton size-classes that are microzooplankton and mesozooplankton. The elemental composition of phytoplankton and non-living organic matter is variable and is prognostically predicted by the model. On the other hand, zooplankton are assumed to be strictly homeostatic, i.e. their stoichiometry is kept constant (e.g., Meunier et al., 2014; Sterner & Elser, 2002). Nutrients uptake and assimilation as well as limitation of growth rate are modeled according to the chain

model of Pahlow and Oschlies (2009). The P quota limits N assimilation which in turns limits phytoplankton growth. The phosphorus to nitrogen ratios of phytoplankton are described based on the potential allocation between P-rich biosynthesis machinery, N-rich light harvesting apparatus, a nutrient uptake component, the carbon stores, and the remainder (Daines et al., 2014; Klausmeier et al., 2004). This allocation depends on the cell size and on the environmental conditions.

Nutrients are delivered to the ocean through dust deposition, river runoff and mobilization from the sediment. The atmospheric deposition if iron is derived from a climatological dust simulation (Tegen and Fung, 1995). The iron from sediment is recognized as a significant source (Johnson et al., 1999; Moore et al., 2004). This iron source is indeed parameterized in PISCES as, basically, a time-constant flux of dissolved iron (2 $\mu mol.m^{-2}.day^{-1}$) applied over the whole sediment surface and modulated depending only on depth. A detailed description of this sedimentary source is presented in Aumont et al. (2015). The initial conditions and biogeochemical fluxes (iron, phosphorus, nitrate, ...) at the boundaries of our domain are extracted from the World Ocean Atlas 2009 (https://www.nodc.noaa.gov/OC5/WOA09/woa09data.html). »

- L137: The reference of Kwiatkowski et al. (subm.) is missing in the bibliography while it is a crucial reference to see the so-called quota version of PISCES model. On the web, this reference cannot be found. Is this paper published now?

The paper is currently in press (at the time we write this review) and reference to this paper has been added.

- L129-141. The description of the model is too succinct. Is there a term of natural mortality on Trichodesmium? What about the trophic interactions between zooplankton and Trichodesmium? Is zooplankton able to graze Trichodesmium or not? From the Table 1 and the parameters inside I can suppose yes but it is not clearly stated and discussed in this section. The two types of zooplankton seem to be able to graze Trichodesmium. No argument is given for this choice. I would like to see a text around this point in the revised ms. Major point to be addressed.

As for the description of the dynamical and biogeochemical models, we decided to add a specific section on the *Trichodesmium* explicit representation in the model. In this description we answer questions about mortality and grazing. To summary, there is a term of natural mortality on Trichodesmium and this term is similar to the other modeled phytoplankton species. The two types of zooplankton are able to graze Trichodesmium , but  we applied two different coefficients for the grazing preference. For microzooplankton, grazing preference is halved to account for *Trichodesmium* toxicity (O'Neil and Romane, 1992).

**« 2.1.3 Trichodesmium compartment**

For the purpose of this study, we implemented in the PISCES-QUOTA version an explicit representation of *Trichodesmium*. Therefore, as already stated, five living compartments are modeled with three phytoplankton groups (nanophytoplankton, diatoms, and *Trichodesmium*) and two zooplankton groups (microzooplankton, and mesozooplankton). Similarly to nanophytoplankton (Equation 1 in Kwiatkowski et al., submitted), the equation of *Trichodesmium* evolution is computed as follows:

$$\frac{\partial Tri_C}{\partial t} = \left(1 - \delta^{Tri}\right)\mu^{Tri} Tri - \zeta_{NO_3}^{Tri} V_{NO_3}^{Tri} - \zeta_{NH_4}^{Tri} V_{NH_4}^{Tri} - m^{Tri}\frac{Tri_C}{K_m + Tri_C}Tri_C \quad \text{(Eq. 1)}$$
$$- sh * w^{Tri} P^2 - g^Z\left(Tri\right)Z - g^M\left(Tri\right)M$$

In this equation, $Tri_C$ is the carbon *Trichodesmium* biomass, and the seven terms on the right-hand side represent respectively growth, biosynthesis costs based on nitrate and ammonium, mortality, aggregation and grazing by micro- and mesozooplankton.

In our configuration, the photosynthesis growth rate of *Trichodesmium* is limited by light, temperature, phosphorus and iron availability. Photosynthesis growth rate of *Trichodesmium* ($\mu^{Tri}$) is computed as follows: $\quad \mu^{Tri} = \mu_{FixN_2} + \mu_{NO_3}^{Tri} + \mu_{NH_4}^{Tri}$ (Eq. 2)

where $\mu_{FixN2}$ denotes growth due to dinitrogen fixation, $\mu^{Tri}_{NO3}$ and $\mu^{Tri}_{NH4}$ represent growth sustained by $NO_3^-$ and $NH_4^+$ uptake, respectively. Moreover, a fraction of fixed nitrogen is released back to seawater, mainly as ammonia and dissolved organic nitrogen, by the simulated *Trichodesmium* compartment. Berthelot et al. (2015) estimated this fraction to be less than 10% when considering all diazotrophs. We set up this fraction at 5% of the total amount of fixed nitrogen. For the other nutrients (i.e. iron and phosphorus), the same fraction is also released.

Dinitrogen fixation is limited by the availability of phosphate, iron and light and is modulated by temperature.

Loss processes are natural mortality, and grazing by zooplankton. Natural mortality is considered to be similar to the other modeled phytoplankton species. Grazing on *Trichodesmium* is rarely described, but it is admitted that *Trichodesmium* represents a poor source of food for zooplankton (O'Neil and Romane, 1992) especially because they contain toxins (Hawser et al., 1992). On the other hand, few species of copepods (mainly mesozooplankton) have been shown to be able to graze on *Trichodesmium* despite the strong concentrations of toxins (O'Neil and Romane, 1992).

For these reasons we applied two different coefficients for the grazing preference by mesozooplankton and microzooplankton (Table 1). For microzooplankton, grazing preference is halved to account for *Trichodesmium* toxicity, and for mesozooplankton the grazing preference is similar to that of the other phytoplankton species. The complete set of equations of *Trichodesmium* is detailed in Appendix 1. Table 1 presents the parameters that differ between nanophytoplankton and *Trichodesmium*.

This setup reproduces dinitrogen fixation through an explicit representation of the *Trichodesmium* biomass (to be compared with often used implicit parameterizations (Assmann et al., 2010; Aumont et al., 2015; Dunne et al., 2013; Maier-Reimer et al., 2005; Zahariev et al., 2008) that link directly environmental parameters to nitrogen fixation without requiring the *Trichodesmium* biomass to be simulated). »

- L142-144: please be careful on the use of the term 'growth rate'. Here, I suppose it is the photosynthesis growth rate and it is not the net growth rate (gains minus losses) of Trichodesmium. Losses are for example grazing or exudation. Please clarify this point.
We replaced 'growth rate' by 'photosynthesis growth rate' for clarity (L185).

- L148-150: What is the form of nitrogen released by Trichodesmium? Is it nitrate and/or ammonium or dissolved organic nitrogen? No information is given on this feature while it is crucial to understand how Trichodesmium can be a source of nitrogen for the plankton community. Moreover, no further information is given on the ability of Trichodesmium to release other element like phosphorus, iron while they are also expected to be constitutive of this planktonic genus. Why are the explicit state variables of the Trichodesmium compartment in fact? To clarify all that points I suggest to add and comment, in this section, a new figure of schematic diagram showing state variables and processes of the Trichodesmium compartment. Major point to be addressed.
It is the belief of the authors that those points have been addressed in the added section « Trichodesmium compartment », already given in response to comments on Line 129 to 141. The idea of adding a diagram has been carefully scrutinized. An informative diagram would have been rather complex and hard to decipher. A more synthetic diagram would be less complex but very close to the already published PISCES diagram. At present, the description in the manuscript has been significantly strengthened and should be clearer than a complex diagram.

In this section, no reference to table 1 is indicated while it is the table of some parameters of the biogeochemical model. More problematic is that all the choices of parameter values are nowhere

discussed in the paper. No references are given in the Table 1. On what criteria have been chosen the values of all these parameters? Is it an arbitrary choice as the exuded fraction of nitrogen by Trichodesmium or from literature? There is an urgent need to justify the values of each parameter presented. I have also some other remarks on the table 1 (see section on table comments hereafter). As another example we don't know why the preference of microzooplankton for Trichodesmium is higher than that of mesozooplankton. I would like to see in the revised version a detailed section on this point. Major point to be addressed.

Table 1 has been modified to include the references from which the parameter values for Trichodesmium have been derived. For nanophytoplankton, parameters values are taken from Aumont et al. (2015). For other parameters, like zooplankton preferences for example, those points have now been addressed in the added section « Trichodesmium compartment », given in response to comments on Line 129 to 141.

Furthermore in the section 2.2 (L159-170), crucial information is lacking about the sediment iron flux while it is a key point debated in the present study. What is the value of this flux and its origin (i.e. literature)? Is it a homogenous spatial flux on the whole model grid or only around some islands? One can imagine for example a decreasing flux from coastal to offshore areas. Neither detailed information nor references to previous works are given. Major point to be addressed.

"The atmospheric deposition of iron is derived from a climatological dust simulation (Tegen and Fung, 1995). The iron from sediment is recognized as a significant source (Johnson et al., 1999; Moore et al., 2004). This iron source is parameterized in PISCES as, basically, a time-constant flux of dissolved iron (2 µmol/m$^2$/day) applied over the whole sediment surface and modulated depending only on depth. A detailed description of this sedimentary source is presented in Aumont et al. (2015)."

Those informations are now given in the "PISCES" section that has been provided to the reviewer in response to its comments on line 128 to 155.

Section 2.3. An important question arises after reading this part of paper. Have the data of OUTPACE cruise been really used to validate the coupled model? This point is not clear to me because the OUTPACE cruise has been carried out in 2015 year while the reference simulation ends 2013 (L160). Several captions of the figures indicate that 'Model values have been sampled at the same location, the same month and the same depth as data'. So I can't understand how it is possible to use the OUTPACE dataset in this paper to validate model output as suggested l175-176 for iron.

Major point to be addressed.

Indeed, this comment is a consequence of our misleading presentation of our model forcing strategy. As explained in a previous comment, our model simulation is climatological. Therefore, the model does not produce any interannual variations. Thus, we compare the model to the observations by sampling our model at the same month and location as the observations. The February to April 2015 OUTPACE observations have then been compared to climatological February to April model outputs. We changed the figures captions to clarify this point:

« Model values have been sampled at the same location, same month (climatological month vs real month), and same depth as the data. »

**5. Results.**

-L193. Phosphate patterns. 'in qualitatively good agreement'. Assertion to be moderated. The model strongly underestimates the areas of high concentrations as within the Costa Rica dome and along the equator.

We have modified the text in the revised version of the manuscript to acknowledge this point : « First, phosphate patterns show modeled values and structures in qualitatively good agreement with observations, despite an underestimation in the areas of high concentrations such as within the Costa Rica dome and along the equator. » (L244-246)

-L197. 'Regions most favourable for Trichodesmium can be defined by temperature within 26-29∘C'. What is the criterion behind this statement? Is it from literature or from a model results (highest biomasses of Trichodesmium)? Author refers these two temperature limits to preferendums in the caption of Fig. 1. How are defined these preferendums?

In the appendix (L528), Equation 3 shows the computation of the temperature limitation term. This statement is based on this equation 3, which shows that the temperature preferendum is reached at ~27°C. Over the range 20-34°C, the limitation function is symmetrical around 27°C, which is the center of the 25-29°C interval. We added a reference to Breitbarth et al., (2007) who proposed this empirical equation (Breitbarth et al., 2007).

-L201-202. I don't agree with the sentence on the good reproduction of seasonal variability of SST along the equator by the model. The 26∘C isotherm migrates by 15∘ eastward in the model from summer to winter but this migration is not observed in the SST field.

We added some text in the revised version of the manuscript to underline this bias: « By contrast, along the equator the mean position of the 25°C isotherm is shifted eastward in the

TRI simulation (~120°W) compared to the observations (~125°W; Fig. 1a vs 1b), but its seasonal displacements are well reproduced except in the South-Eastern Pacific. » (L254-256).

-L205. Please indicate the type of statistical test (and probability) used to prove the result of 'no statistical differences'.

We added in the text these informations:

« The median value as well as the dispersion of the iron surface concentrations over the tropical Pacific, are displayed for both the data and the model in Figure 2a. Mann-Whitney test reveals that these two normalized distributions are not significantly different (p-value = 0.26). » (L260-263).

-L214. 'with mean values higher than 0.3 mgChl m-3'.

The text has been amended (L271).

-L221. 'Those localized chlorophyll . . . effect'. This sentence should be in discussion.

-L224-225. 'TRI simulation thus appears. . .' This sentence should be in discussion.

We moved these two sentences in the discussion section.

-L232. 'SPG'. Acronym not defined.

We modified the text to define the acronym:

« This bias in the model could be explained by the overestimated iron concentrations in the South Pacific Gyre (SPG). » (L288).

-L236-253. There is neither clear explanation nor associated analysis why the numerical N2 fixation rates are compared with data over two different integration layers (Fig. 4).

We added those sentences to explain the reasons of these two integration layers:

« Some areas are sampled only in the surface layer (0-30m) while others have been sampled deeper. This non-homogeneous sampling may alter the distribution of the N2 fixation rates and undermine the comparison with model outputs. To assess and overcome this sampling bias we compared the observations with simulated N2 fixation rates over two different integration layers (0-30m and 0-150m). » (L294-296).

-L244. Same as my previous remark on the use of OUTPACE iron data in the validation step of model (section 2.3.). How OUTPACE N2 fixation data '(Bonnet et al., this issue)' can be used to validate the model as the simulation ends 2013?

As previously mentioned, the model simulations are climatological. Thus, data comparisons are

made vs. seasonal climatologies.

-L247. 'In general, ... compared to data'. This sentence is vague and then confusing. Is an overestimation on the whole modelled domain, or only in one sector especially? Is this statement applicable for the rates depth-integrated 0-150m or 0-30m, or both?

The text has been modified to make this sentence clearer:

« On the whole modeled domain and for both integration layers, dinitrogen fixation rates are overestimated by ~70% in TRI compared to the data. » (L305-306).

-L256. The Figure 6 should be numbered 5 instead 6 because it follows the description of Figure 4.

Figures 5,6,7 are now 7,5,6 respectively.

-L261. 'PNG'. Acronym not defined.

We now defined this acronym in the following sentence:

« Maximum values are located in the South West Pacific (around Vanuatu archipelago, New Caledonia, Fiji, and Papua New Guinea (PNG)) and around Hawaii, where they reach 0.06 mg Chl.m$^{-3}$. » (L284).

-L272. Why does Author indicate the term 'not shown' for the simulated N2 fixation rates and Trichodesmium biomass as they are presented on Figures 5 and 7, respectively?

Indeed, this is shown in figures 6 and 7 (with the updated numbering) which are now properly referenced (L330).

-L275. 'Figure (6 a,b)' instead Figure (7a,c).

Text has been modified (L333)

-L284. Same remark as L205.

The requested information has been added in the text (L341-343).

« They proved that vertically integrated dinitrogen fixation rates are statistically significantly (one-way ANOVA, $p<0.01$) lower from November to March (less than 200 µmol N m$^{-2}$.d$^{-1}$) than from April to October (about $263 \pm 147$ µmol N.m$^{-2}$.d$^{-1}$) as highlighted in Figure 7a (blue dots). »

-L286. 'Figure 7a' instead 'Figure 5a'.

The numbering has been corrected (L344)

-L303. Please replace 'in the sampling' by 'in the field observations'.

We have replaced 'in the sampling' by 'if sampled at the observed stations' (L 361-362).

-L306-308. What about the other factors (as grazing or natural mortality if existing)? What is the type of analysis exactly, leading to the conclusion 'the seasonal variability is mainly controlled by primary production'? Please replace the term 'by primary production' by 'by the levels of primary production'.

The other loss terms such as natural mortality and grazing have been analyzed in a similar manner to what is shown in Figure 7. This analysis showed that they do not control the shape of the seasonal cycle. They rather play a role in its amplitude, not in its shape. That's the reason why we have not shown the analysis in the manuscript. We have modified the text to discuss this point in the manuscript :

« To further investigate the mechanisms that drive the seasonal variability of *Trichodesmium* in the Pacific, we examined the factors that control Trichodesmium abundance in the TRI simulation (not shown). This decomposition shows that the physical terms (advection and mixing) are negligible compared to biological terms. In addition, the seasonal cycles of grazing and mortality are in phase with the production terms but their sign is opposite. In conclusion, this analysis indicates that this seasonal variability is mainly controlled by the levels of primary production, the others terms of tracer evolution dampe its amplitude but do not change its shape. » (L365-370).

You can see this analysis in Fig. Supp 3.

Furthermore, following your suggestion, the term 'primary production' has been replaced by 'by the level of primary production' (L369).

-L312-316. Would it be possible to see (in a new table for example) a synthesis of the modelled values of Trichodesmium growth rates of and a comparison of them with those observed in the field if existing or in lab experiments.

This would have required to add the *Trichodesmium* growth rates as outputs of our simulations, which is not the case, and would require a significant amount of time to be processed (all simulations would have to be rerun). Furthermore, the growth rates of Trichodesmium simulated in our model do exhibit a strong spatial and temporal variability which makes the presentation in a table challenging. A comparison with in situ data is difficult to perform because appropriate data are very scarce in the literature to our knowledge. Most data focus on nitrogen fixation rates and more occasionally, on net population growth rates, rather than on in situ photosynthetic growth rates. Many more observations based on laboratory experiments are available but they correspond to

controlled conditions which are difficult to compare to the actual conditions simulated by our model.

-L324-326. 'Indeed, . . . temperature'. This sentence should be placed in discussion. Furthermore, this sentence is highly debatable. Please be cautious with the concept of 'ocean dynamics mainly 1D'! Is this feature really achieved anywhere in the ocean? Can the Author firmly prove this assertion in the simulations presented in this study?

Indeed, this reference to a 1D- ocean was mentioned to stress that the seasonal enhanced vertical mixing is bringing to the euphotic zone iron-replete waters that are cold water masses in our boxes, and hence have antagonist effects on the Trichodesmium biomasses. Thus, we proposed the following new wording which insists on the seasonal enhanced mixing without referring to a 1D-ocean:

« Indeed, nutrients and iron inputs brought to the euphotic zone by the seasonally enhanced vertical mixing are counter-balanced by the related inputs of cold water masses. » (L386-388)

However to prove our assertion we performed a new analysis (Fig. Supp 3). This figure show the decompostion of all physics terms for iron tracer in south pacific (red box, integrated over the top 150m of the ocean, and averaged on the region; Fig 1c). We find that vertical mixing dominates the other terms (advection and horizontal mixing) all the year except in November and December. Moreover, the correlation coefficient between vertical mixing and the iron time rate of change is 0.95, therefore at the 0 order the vertical mixing controls the iron time rate of change. Obviously for this tracer it is also necessary to take into account the evolution of biological terms.

**6. Discussion.**

-L369. I would like to see a clear definition of the term 'bio-available nitrogen'. Is it dissolved inorganic forms of nitrogen and/or organic forms also?

This terms refers to the nitrogen forms that can be taken up by phytoplankton. In our model, that is nitrate and ammonium. This definition has been appended in the manuscript.

« *Trichodesmium* also releases a fraction of the recently fixed $N_2$ as bio-available nitrogen (in our model, Trichodemium releases ammonia and dissolved organic nitrogen, but only ammonia is directly bio-available). » (L436-438)

-L393. 'to that of' instead of 'than'.

The text has been corrected (L449).

-L400-403. What is the actual reason for a better modelling of N2 fixation by using an explicit

representation of this process in the model? At the place of the manuscript one can expect a deeper analysis of the results. A thorough comparison of the two types of formulation could be lead to explain clearly why using the explicit formulation is an improvement. Is it due, for example, to the inclusion of Fe internal quota in the formulation of Trichodesmium photosynthesis growth rate? Major point to be addressed.

We agree with the reviewer that a more detailed analysis of the factors that explain the differences between the implicit and explicit formulations would be interesting. However, such comparison is far from being easy to carry out. The main reason for this difficulty is that the explicit formulation includes multiple non-linear interactions (grazing, growth …) that are not represented in the implicit parameterization. These non-linearities are probably an explanation to the better behavior of the explicit model. They allow the model to predict blooms of *Trichodesmium* and as explained in the manuscript, the main improvement brought by the explicit representation of *Trichodesmium* is the larger biomass, especially near the islands.

To deepen this comparison we decided, following the comment of the other reviewer, to add 2 analysis, in order to compare the $N_2$ fixation rate (Fig. Supp 1) and the carbon export (Fig Supp 2) in TRI an TRI_imp simulations.

Figure supp. 1 represents the carbon export (under the euphotic layer, in $\mu mol\ N.m^{-2}d^{-1}$ ) comparison and the figure supp. 2 represents the $N_2$ fixation rate comparison (integrated over top to 150m, panel a in $mmol\ C.\ s^{-2}.d^{-1}$ and panel b in percentage). We observe a carbon export greater in TRI simulation, the average across the Pacific of this difference is $0.1mmol\ C.m^{-2}.d^{-1}$ or 4 %, and in LNLC regions the increase varies between 6 and 10 %. The $N_2$ fixation rates are greater in TRI simulation except in the warm pool, in the equatorial upwelling, and in Peru upwelling.

**7. Appendix.**

In this section all the equations presented should be numbered for clarity.

We numbered all the equations.

-L477. I suggest to write 'phosphorus or iron' rather than 'phosphorus and iron'.

We replaced « phosphorus or iron» by «phosphorus and iron».

-L479-480. On which basis (literature, experimental works?) the equations of phosphorus and iron limitations have been stated? Are they new formulations? What physiological processes drive the choice of this formulation? I would like to see information on that point in the revised ms.

As described in PISCES section, the formulations stems from the chain model of Pahlow and Oschlies (2009):

« Nutrients uptake and assimilation as well as limitation of growth rate are modeled according to the chain model of Pahlow and Oschlies (2009). The P quota limits N assimilation which in turns limits phytoplankton growth. The phosphorus to nitrogen ratios of phytoplankton are described based on the potential allocation between P-rich biosynthesis machinery, N-rich light harvesting apparatus, a nutrient uptake component, the carbon stores, and the remainder (Daines et al., 2014; Klausmeier et al., 2004). This allocation depends on the cell size and on the environmental conditions ».

-L480. 'Nutrient quota for Fe and phosphorus' rather than 'Nutrient quota for Fe and PO4'.
We replaced « Nutrient quota for Fe and PO4 » Nutrient quota for Fe and phosphorus » by « Nutrient quota for Fe and phosphorus ».

-L492. Equation of Trichodesmium growth rate if iron limiting. Same remark as for L479- 480. Why the N2 fixation growth rate in case of iron limitation is modelled in this way? It is very difficult to evaluate this formulation without explanation! Furthermore, please be careful in using the term ï ¿A ¿a that can be confused with the term ï ¿A ¿aI (initial slope of P. vs. I.). It is not clear to me if the value of µTRIMAX is of 0.25 d-1? If yes, please clearly indicate in the table of parameters (Table 1) its value and at l473. The term LI is used while undefined.
We changed the organization of this section, and we rewrote the equations differently in more details in order to facilitate their understanding. µTRIMAX is the maximum observed growth rate, and this value comes from the experiences of Breitbarth et al. (2007).

-L518. While the limiting function by temperature is defined previously (L471, LT), the limiting function by light is not presented and it deserves to show it. What is the exact form of the term LI defined by the author?
It is now specified in the manuscript that the light limitation remains similar between the *Trichodesmium* and the nanophytoplankton groups. This limitation is fully detailed in Aumont et al. (2015): « Visible light is split into three wavebands: blue (400-500 nm), green (500-600nm) and red (600-700nm). For each waveband, the chlorophyll-dependent attenuation coefficients are fitted to the coefficients computed from the full spectral model of Morel (1988) (as modified in Morel and Maritorena (2001)) assuming the same power-law expression. At the sea surface, visible light is split equally between the three wavebands ».

-L521. Please be cautious in using the terms of 'new and regenerated production' in this context. The growth rates of Trichodesmium based on nitrate and ammonium are not strictly speaking new

and regenerated productions, respectively. Please reconsider this sentence and formulate your idea with accuracy.

This sentence has been removed, and generally the appendix section has been fully reworked to answer your comments.

Below the new appendix section:

« *Trichodesmium* preferentially fixes dinitrogen at temperature between 20-34°C (Breitbarth et al., 2007). The temperature effect on the growth rate is modeled using a 4th order polynomial function (Ye et al., 2012):

$$L_{Tri}^T = \frac{2,32.\,10^{-5} \times T^4 - 2,52.10^{-3} \times T^3 + 9,75.\,10^{-2} \times T^2 - 1,58 \times T + 9.12}{0.25} \quad \text{(Eq. 3)}$$

where 0.25d$^{-1}$ is the maximum observed growth rate (Breitbarth et al., 2007). Hence, at 17°C the growth rate is zero and maximum growth rate is reached at 27°C. The *Trichodesmium* light limitation is similar to nanophytoplankton (Aumont et al. (2015)).

From equation 2, we distinguish 2 cases for the growth rate due to nitrogen fixation.

if phosphorus is limiting equation 2 becomes :

$$\mu_{Fix} = \mu_{max}^{Tri}.\,L_I^{Tri}.\,L_P^{Tri} - \left(\mu_{NO_3}^{Tri} + \mu_{NH_4}^{Tri}\right) \quad \text{(Eq. 4a)} \quad \text{with} \quad L_P^{Tri} = min\left(1, max\left(0, \frac{\left(\theta^P - \theta_{min}^P\right) \times \theta_{max}^P}{\left(\theta_{max}^P - \theta_{min}^P\right) \times \theta^P}\right)\right) \quad \text{(Eq. 4b)}$$

if iron is limiting :

$$\mu_{Fix} = \mu_{max}^{Tri}.\,L_I^{Tri}.\,L_{Fe}^{Tri} - \left(\mu_{NO_3}^{Tri} + \mu_{NH_4}^{Tri}\right) \quad \text{(Eq. 5a)} \quad \text{with} \quad L_{Fe}^{Tri} = min\left(1, max\left(0, \frac{\left(\theta^{Fe} - \theta_1^{Fe}\right).\,\theta_{opt}^{Fe}}{\left(\theta_{opt}^{Fe} - \theta_0^{Fe}\right).\,\theta^{Fe}}\right)\right) \quad \text{(Eq. 5b)}$$

In equation 4b, $\theta^{Fe}_1$ and $\theta^{Fe}_0$ are computed as follows :

$$\theta_1^{Fe} = \theta_0^{Fe} + \alpha.\,\mu_{Fix} \quad \text{(Eq. 6a),} \qquad \theta_0^{Fe} = \theta_{min}^{Fe} + m \quad \text{(Eq.6b), and} \qquad \alpha = \frac{1}{\beta} \quad \text{(Eq. 6c)}$$

$\theta^{Nutrients}$ represents the nutrient quota for Fe and phosphorus (i.e, the ratio between iron and carbon concentrations in *Trichodesmium*, for instance). $\theta_{min}^P$, and $\theta_{opt}^{Nut}$ are constants, whereas $\theta^{Nutrients}$ varies with time. The mimimum of $L_{Fe}^{Tri}$ and $L_P^{Tri}$ defines the limiting nutrient. $L_I$ is the limiting function by temperature and light.

m represents the difference between the maintenance iron (i.e, the intracellular Fe:C present in the cell at zero growth rate) under diazotrophic growth and growth on ammonium (Kustka et al., 2003). β is the marginal use efficiency and equals the moles of additional carbon fixed per additional mole of intracellular iron per day (Raven, 1988; Sunda and Huntsman, 1997).

The demands for iron in phytoplankton are for photosynthesis, respiration and nitrate/nitrite reduction. Following Flynn and Hipkin (1999), we assume that the rate of synthesis by the cell of new components requiring iron is given by the difference between the iron quota and the sum of the iron required by these three sources of demand, which we defined as the actual minimum iron quota:

$$\theta_{min}^{Fe} = \frac{0.0016}{55.85}\theta_{Tri}^{Chl} + \frac{1.2110^{-5}\times 14}{55.85\times 7.625}L_P^{Tri} + \frac{1.1510^{-4}\times 14}{55.85\times 7.625}L_{NO_3}^{Tri} \quad \text{(Eq. 7)}$$

In this equation, the first right term corresponds to photosynthesis, the second term corresponds to respiration and the third term estimates nitrate and nitrite reduction. The parameters used in this equation are directly taken from Flynn and Hipkin (1999).

**8. Bibliography.**

Please check carefully this section. Many typos and different formats.

We have checked carefully this section, and corrected many typos and different formats.

**9. Tables.**

Table 1. Major points to be addressed. Why are only presented the parameters of Trichodesmium and nanophytoplankton? Are the parameters of other living biomass compartments remained unchanged (what is the reference in which the unchanged parameters can be found?)? If yes, why those of nanophytoplankton only have been changed? I would like to see in the revised ms explanations on this point. The column 'Name in the code' is useless (technical details) but adding a column with references for each parameter is essential. The important parameter µTRIMAX is missing in the table. Please check carefully the units of each parameter. According to the definition of ï¿Ac given in Appendix (L501-502), its unit cannot be in d-1. Replace the term 'excretion' ´ by 'exudation' for parameters 'rTri' and 'rI'.

We presented *Trichodesmium* and nanophytoplankton parameters so that we can compare them. The nanophytoplankton and diatoms parameters remain unchanged from Kwiatkowski et al. (2018). Now, this table only displays the *Trichodesmium* parameters that differ from those of nanophytoplankton. The column 'name in the code' has been removed. We changed the unit of marginal use efficiency.

**Table 1 : Models parameters for Trichodemium and nanophytoplakton.**

| Parameters | Symbol | Unity | Value | Reference |
|---|---|---|---|---|
| Maximum growth rate for Tricho. | $\mu^{Tri}_{max}$ | $d^{-1}$ | 0.25 | Breitbarth et al. (2007) |
| Maximum growth rate for Nano. | $\mu^{Nano}_{max}$ | $d^{-1}$ | 1.0 | |
| Initial slope P-I tricho | $\alpha I$ | $(W.m^{-2})^{-1} .d^{-1}$ | 0.072 | Breitbarth et al. (2008) and Hood et al. (2002) |
| Initial slope P-I nano | $\alpha I$ | $(W.m^{-2})^{-1} .d^{-1}$ | 2.0 | |
| Microzoo preference for Tricho. | pItri | - | 0.5 | |
| Microzoo preference for nano | pIP | - | 1.0 | |
| Maximum Fe/C in Tricho. | $\theta Fe,Trimax$ | $mol\ Fe.(mol\ C)^{-1}$ | $1.10^{-4}$ | Kustka et al. (2003) |
| Maximum Fe/C in nanophyto | $\theta Fe,Imax$ | $mol\ Fe.(mol\ C)^{-1}$ | $4.10^{-5}$ | |
| Maintenance iron | m | $mol\ Fe.(mol\ C)^{-1}$ | $1.4.10^{-5}$ | Kustka et al. (2003) |
| Marginal use efficiency | $\beta$ | $mol\ C.(mol\ Fe)^{-1}.day^{-1}$ | $1.4.10^{-4}$ | Kustka et al. (2003) |

**10. Figures captions.**

Fig.1. Typo: 'preferendum'.

The modification has been done.

Fig. 2. Typo '0-150m'.

The modification has been done.

Fig. 5b. 'The green curve is the average of the seasonal cycle. . .' This sentence is not clear to me. How is this average built exactly? This is no more clearly explained in the corresponding section (l300-301).

The sentence has been changed in the figure caption to: « The green curve represents the seasonal cycle computed from model outputs sampled during the same month at the same location than data, which have then be spatially averaged. »

Table 1 : Models parameters for Trichodemium and nanophytoplakton.

| Parameters | Symbol | Unity | Value | Reference |
|---|---|---|---|---|
| Maximum growth rate for Tricho. | $\mu^{Tri}_{max}$ | $d^{-1}$ | 0.25 | Breitbarth et al. (2007) |
| Maximum growth rate for Nano. | $\mu^{Nano}_{max}$ | $d^{-1}$ | 1.0 | |
| Initial slope P-I tricho | $\alpha I$ | $(W.m^{-2})^{-1} .d^{-1}$ | 0.072 | Breitbarth et al. (2008) and Hood et al. (2002) |
| Initial slope P-I nano | $\alpha I$ | $(W.m^{-2})^{-1} .d^{-1}$ | 2.0 | |
| Microzoo preference for Tricho. | pItri | - | 0.5 | |
| Microzoo preference for nano | pIP | - | 1.0 | |
| Maximum Fe/C in Tricho. | $\theta Fe,Trimax$ | $mol\ Fe.(mol\ C)^{-1}$ | $1.10^{-4}$ | Kustka et al. (2003) |
| Maximum Fe/C in nanophyto | $\theta Fe,Imax$ | $mol\ Fe.(mol\ C)^{-1}$ | $4.10^{-5}$ | |
| Maintenance iron | m | $mol\ Fe.(mol\ C)^{-1}$ | $1.4.10^{-5}$ | Kustka et al. (2003) |
| Marginal use efficiency | $\beta$ | $mol\ C.(mol\ Fe)^{-1}.day^{-1}$ | $1.4.10^{-4}$ | Kustka et al. (2003) |

Table 2 : List and description of the different experiments.

[revised manuscript text omitted]

Fig. Supp. 1 : Difference of fluxes of carbon export at euphotic layer between TRI simulation and TRI_imp simulation in mmol C.m$^{-2}$.d$^{-1}$  (a) and in percentage (b).

Fig. Supp. 2 : Depth-integrated (from 0 to 150m) rates of nitrogen fixation ($\mu$mol N.m$^{-2}$d$^{-1}$) in a) TRI simulation and b) TRI_imp simulation.

Fig. Supp. 3 : Physic balance of depth-integrated (from 0 to 150m) rate of iron concentrations (in mol Fe.m$^{-2}$.d$^{-1}$) averaged on south Pacific (red box, Fig. 1).

Fig. Supp. 4 : Seasonal cycle of Trichodesmium rate change (in mol C.m$^{-2}$.s$^{-1}$) in the South Pacific.

[Figure]

Figure 1

[Figure]

[Figure]

Figure 2

[Figure]

Figure 3

[Figure]

[Figure]

Figure 4

[Figure]

[Figure]

Figure 5

[Figure]

Figure 6

[Figure]

Figure 7

[Figure]

Figure 8

[Figure]

Figure 9

[Figure]

Figure 10

[Figure]

[Figure]

Figure Supp. 1

[Figure]

Figure Supp. 2

[Figure]

FERRET Ver. 6.95
NOAA/PMEL TMAP
17-APR-2018 09:34:35

LONGITUDE : 160E to 130W (XY ave)
LATITUDE : 25S to 14S (XY ave)
DEPTH (m) : 0 to 150 (integrated)

DATA SET: Z_roms_diags_avg_TS_climato

PAC_1

Sum of all physics terms
averaged horizontal advection terms
averaged vertical advection terms
averaged horizontal mixing term
averaged vertical mixing term

Figure Supp. 3

[Figure]

Figure Supp. 4

---

## Author Comment (AC2) · 23 Apr 2018

The manuscript describes model simulations without and with two different parameterizations of nitrogen fixation in the tropical Pacific. Results are compared against observations in the ocean's surface layer, and the degree of realism of the two parameterizations employed is discussed. Inferences are made about the role of diazotrophic nitrogen fixation compared to primary production by ordinary phytoplankton.

Overall, the topic is scientifically very interesting and I found the title and also the abstract very promising, but was then disappointed by the material presented in the manuscript (and the often poor way it was presented) for reasons I will explain below. I am afraid I cannot recommend publication of the manuscript in its present form and think that a very major rewrite and additional and thorough analysis is required. This is beyond what I would normally consider as major revision (and would therefore recommend reject and resubmission). As the issue is tricky with special issues, and because the scientific topic is really interesting and it would be a missed opportunity of not analysing this very carefully, I'm still OK with recommending major submission, but want to stress that 'major' should be taken very seriously.

**Legend for the review :**

In blue our answers.

**Response to general comments:**

To answer your comments we have completely rewritten the method and appendix section. The result section has also been reworked so that the speech is clearer and more precise. We have also strengthened the introduction by more accurately detailing the state of the art of nitrogen fixation in biogeochemical models.

**1. It is impossible to fully understand what has been done**

The explicit description of N2 fixation by Trichodesmium is provided in the Appendix. I tried hard to understand it, but admit that I failed. There may be typos or unexplained Fe terms (e.g., what is L N T ri in line 492? Why are there two different definitions of L T ri , lines 479 and 499?). It does not help, that the notation in table 1 seems to be different from the one in the appendix. There are also steps that are not explained or justified. For example line 483 - why is this procedure applied to Fe but not to P? This makes it impossible to understand what has been done and why. There are other models of diazotrophs out in the literature. How does your model relate to these? Why have you developed a new one (is it new?)? To be useful to the scientific community, this has to be presented in much more detail and put into relation to the existing literature.

Following your comments, we have significantly modified the parts presenting our set up, the context and the description of the explicit representation of *Trichodesmium*.

A paragraph about the different models used in the literature has been added in the introduction section to contextualize our study:

« Numerical models have also been used as they allow to overcome the scarcity of observations that may limit the implementation of the two previous approaches (Aumont et al., 2015; Bissett et al., 1999; Dutkiewicz et al., 2012; Keith Moore et al., 2006; Krishnamurthy et al., 2009; Monteiro et al., 2011; Moore et al., 2013; Tagliabue et al., 2008). They can notably be used to investigate the spatial and temporal variability of $N_2$ fixation and to study how and which environmental factors control this process. In these models, $N_2$ fixation has been implemented in various ways. Some models use implicit parameterizations (Bisset et al., 1999; Maier-Reimer and Kriest, 2005; Assmann et al., 2010; Aumont et al., 2015) to derive $N_2$ fixation from environmental conditions (mainly nitrate, phosphate and iron concentrations, temperature and light). Alternatively, other models rely on the explicit descriptions of diazotrophs (Moore et al., 2004; Dunne et al., 2013) that have mainly been developed from the knowledge derived from laboratory culture experiments focused on *Trichodesmium* sp. (Fennel et al., 2001; Hood et al., 2001; Moore et al., 2001). Noticeably, several modeling studies have been especially focused on the role of iron in controlling the distribution of diazotrophs and $N_2$ fixation (Keith Moore et al., 2006; Krishnamurthy et al., 2009; Moore et al., 2004; Tagliabue et al., 2008). Indeed, a realistic representation of marine iron concentrations has been stressed as a key factor to adequately simulate the habitat of diazotrophs (Monteiro et al., 2011; Dutkiewicz et al., 2012). »

The most relevant informations of the explicit modelisation of the *Trichodesmium* compartment have been added to the manuscript within a specific section:

**« 2.1.3 The *Trichodesmium* compartment**

For the purpose of this study, we implemented an explicit representation of Trichodesmium in the PISCES-QUOTA version. Therefore, as already stated, five living compartments are modeled including three phytoplankton groups (nanophytoplankton, diatoms, and *Trichodesmium*) and two zooplankton groups (microzooplankton, and mesozooplankton). Similarly, to nanophytoplankton (Equation 1 in Kwiatkowski et al., submitted), the equation of Trichodesmium evolution is computed as follows:

$$\frac{\partial Tri_C}{\partial t} = \left(1 - \delta^{Tri}\right)\mu^{Tri} Tri - \zeta^{Tri}_{NO_3} V^{Tri}_{NO_3} - \zeta^{Tri}_{NH_4} V^{Tri}_{NH_4} - m^{Tri}\frac{Tri_C}{K_m + Tri_C} Tri_C$$
$$- sh * w^{Tri} P^2 - g^Z(Tri)Z - g^M(Tri)M$$

(Eq. 1)

In this equation, $Tri_C$ is the carbon Trichodesmium biomass, and the seven terms on the right-hand side represent respectively growth, biosynthesis costs based on nitrate and ammonium, mortality, aggregation and grazing by micro- and mesozooplankton.

In our configuration, the photosynthesis growth rate of *Trichodesmium* is limited by light, temperature, phosphorus and iron availability. Photosynthesis growth rate of *Trichodesmium* ($\mu^{Tri}$) is computed as follows: $\mu^{Tri} = \mu_{FixN_2} + \mu^{Tri}_{NO_3} + \mu^{Tri}_{NH_4}$ (Eq. 2)

where $\mu_{FixN2}$ denotes growth due to $N_2$ fixation, $\mu^{Tri}_{NO3}$ and $\mu^{Tri}_{NH4}$ represent growth sustained by $NO_3^-$ and $NH_4^+$ uptake, respectively. Moreover, a fraction of fixed nitrogen is released back to seawater, mainly as ammonia and dissolved organic nitrogen, by the simulated Trichodesmium compartment. Berthelot et al., (2015) estimated this fraction to be less than 10% when considering all diazotrophs. We set up this fraction at 5% of the total amount of fixed nitrogen. For the other nutrients (i.e. iron and phosphorus), the same fraction is also released.

$N_2$ fixation is limited by the availability of phosphate, iron and light and is modulated by temperature.

Loss processes are natural mortality, and grazing by zooplankton. Natural mortality is considered to be similar to the other modeled phytoplankton species. Grazing on *Trichodesmium* is rarely described, but it is admitted that *Trichodesmium* represents a poor source of food for zooplankton (O'Neil and Romane, 1992) especially because they contain toxins (Hawser et al., 1992). On the other hand, few species of copepods have been shown to be able to graze on *Trichodesmium* despite the strong concentrations of toxins (O'Neil and Romane, 1992). For these reasons we applied two different coefficients for the grazing preference by mesozooplankton and microzooplankton (Table 1). For microzooplankton, grazing preference is halved to account for Trichodesmium toxicity, and

for mesozooplankton the grazing preference is similar to that of the other phytoplankton species. The complete set of equations of *Trichodesmium* is detailed in Appendix 1. Table 1 presents the parameters that differ between Nanophytoplankton and Trichodesmium.

This setup reproduces $N_2$ fixation through an explicit representation of the Trichodesmium biomass (to be compared with often used implicit parameterizations (Assmann et al., 2010; Aumont et al., 2015; Dunne et al., 2013; Maier-Reimer et al., 2005; Zahariev et al., 2008)) that links directly environmental parameters to $N_2$ fixation without requiring the Trichodesmium biomass to be simulated). »

In addition to those changes within the manuscript, we significantly modified the appendix :

« *Trichodesmium* preferentially fixes di-nitrogen at temperature between 20-34°C (Breitbarth et al., 2007). The temperature effect on the growth rate is modeled using a 4th order polynomial function (Ye et al., 2012):

$$L_T^{Tri} = \frac{2,32.\,10^{-5} \times T^4 - 2,52.\,10^{-3} \times T^3 + 9,75.\,10^{-2} \times T^2 - 1,58 \times T + 9.12}{0.25} \quad \text{(Eq. 3)}$$

where 0.25d$^{-1}$ is the maximum observed growth rate (Breitbarth et al., 2007). Hence, at 17°C the growth rate is zero and maximum growth rate is reached at 27°C. The Trichodesmium light limitation is similar to nanophytoplankton (Aumont et al. (2015)).
From equation 2, we distinguish 2 cases for the growth rate due to $N_2$ fixation.
if phosphorus is limiting the equation 2 becomes :

$$\mu_{Fix} = \mu_{max}^{Tri} . L_I^{Tri} . L_P^{Tri} - \left( \mu_{NO_3}^{Tri} + \mu_{NH_4}^{Tri} \right) \quad \text{(Eq. 4a) with} \quad L_P^{Tri} = min\left(1, max\left(0, \frac{\left(\theta^P - \theta_{min}^P\right) \times \theta_{max}^P}{\left(\theta_{max}^P - \theta_{min}^P\right) \times \theta^P}\right)\right) \quad \text{(Eq. 4b)}$$

if iron is limiting :

$$\mu_{Fix} = \mu_{max}^{Tri} . L_I^{Tri} . L_{Fe}^{Tri} - \left( \mu_{NO_3}^{Tri} + \mu_{NH_4}^{Tri} \right) \quad \text{(Eq. 5a) with} \quad L_{Fe}^{Tri} = min\left(1, max\left(0, \frac{\left(\theta^{Fe} - \theta_1^{Fe}\right) \times \theta_{opt}^{Fe}}{\left(\theta_{opt}^{Fe} - \theta_0^{Fe}\right) \times \theta^{Fe}}\right)\right) \quad \text{(Eq. 5b)}$$

In equation 4b, $\theta^{Fe}_1$ and $\theta^{Fe}_0$ are computed as follows :

$$\theta_1^{Fe} = \theta_0^{Fe} + \alpha . \mu_{FixN_2} \quad \text{(Eq. 6a),} \quad \theta_0^{Fe} = \theta_{min}^{Fe} + m \quad \text{(Eq.6b), and} \quad \alpha = \frac{1}{\beta} \quad \text{(Eq. 6c)}$$

$\theta^{Nutrients}$ represents the nutrient quota for Fe and phosphorus (i.e, the ratio between iron and carbon

concentrations in *Trichodesmium*, for instance). $\theta_{min}^P$, and $\theta_{opt}^{Nut}$ are constants, whereas $\theta^{Nutrients}$ varies with time. The mimimum of $L^{Tri}_{Fe}$ and $L^{Tri}_P$ defines the limiting nutrient. $L_I$ is the limiting function by temperature and light.

m represents the difference between the maintenance iron (i.e, the intracellular Fe:C present in the cell at zero growth rate) under diazotrophic growth and growth on ammonium (Kustka et al., 2003). β is the marginal use efficiency and equals the moles of additional carbon fixed per additional mole of intracellular iron per day (Raven, 1988; Sunda and Huntsman, 1997).

The demands for iron in phytoplankton are for photosynthesis, respiration and nitrate/nitrite reduction. Following Flynn and Hipkin (1999), we assume that the rate of synthesis by the cell of new components requiring iron is given by the difference between the iron quota and the sum of the iron required by these three sources of demand, which we defined as the actual minimum iron quota:

$$\theta_{min}^{Fe} = \frac{0.0016}{55.85}\theta_{Tri}^{Chl} + \frac{1.21 \cdot 10^{-5} \times 14}{55.85 \times 7.625} L_P^{Tri} + \frac{1.15 \cdot 10^{-4} \times 14}{55.85 \times 7.625} L_{NO_3}^{Tri} \quad \text{(Eq. 7)}$$

In this equation, the first right term corresponds to photosynthesis, the second term corresponds to respiration and the third term estimates nitrate and nitrite reduction. The parameters used in this equation are directly taken from Flynn and Hipkin (1999).

The authors claim that implicit parameterizations of $N_2$ fixation are often used in biogeochemical models (line 32, line 154, in the final sentence of the manuscript they even say 'more commonly'), but do not provide a single reference to support this claim. I think this strong statement that is used and certainly requires references and also a detailed description of this implicit parameterisation in order to allow the reader to understand some of the results (see below), and possibly repeat what has been done here.

We added references (L187-188) for the models that we know use implicit parameterization of $N_2$ fixation. Martinez-Rey (2015) presents in his thesis a list of the parameterizations of $N_2$ fixation used in the biogeochemical models embedded in the CMIP5 models. On 10 CMIP5 models, 2 biogeochemical models use an explicit description of $N_2$ fixation, 6 use an implicit formulation of $N_2$ fixation and 2 have no representation of $N_2$ fixation.

As already stated, the introduction has been modified and we followed the reviewer's recommendation and replaced the two sentences referring to numerical models (L119 to 121 in the submitted manuscript) by the paragraph already given at the beginning of this review.

We also added the main characteristics of the implicit $N_2$ fixation scheme used in our study in the

section « experimental setup » :

« In a third experiment "N2_imp", the explicit dinitrogen fixation module is replaced by the implicit parameterization described in Aumont et al. (2015) where fixation depends directly on water temperature, nitrogen, phosphorus and iron concentrations and light (no nitrogen fixers are simulated). »

We did not feel that more details were needed as a specific description has already been published in Aumont et al., (2015).

The set-up of the physical model is unclear as well. line 111 states that it is based on a nested version. Is there a nested version used here? If so, what is the parent and what the child model? Then, in line 116 open boundary conditions are introduced. Do these replace the nesting? What does the sentence in line 118 mean "The use of similar ROMS configurations. . .is validated. . ."?

Indeed, the wording used in this section was confusing and some information about our simulations were missing. We use a regional model with open boundaries. Thus, there is no nest in this configuration. The confusing sentence referring to the « nested version » have been removed and we now only refer to the ROMS-AGRIF version of the model. The sentence « The use of similar ROMS configurations. . .is validated. . . » means that some validation of the physical conditions of the South Pacific region produced by this ROMS configuration (e.g vertical resolution, mixed active/passive scheme, turbulent vertical mixing parameterization) has been already published (Jullien et al., 2012, 2014; Marchesiello et al., 2010).

The whole section has been extensively modified in order to take your comments into account:

**« 2.1.1 ROMS**

In this study, we used a coupled dynamical-biogeochemical framework based on the regional ocean dynamical model ROMS (Regional Oceanic Modeling System, (Shchepetkin and McWilliams, 2005)) and the state of the art biogeochemical model PISCES (Pelagic Interactions Scheme for Carbon and Ecosystem Studies). The ocean model configuration is based on the ROMS-AGRIF (Penven et al., 2006) informatic code and covers the tropical Pacific region [33°S-33°N;110°E-90°W]. It has 41 terrain-following vertical levels with 2-5 m vertical resolution in the top 50 meters of the water column, then 10-20 m resolution in the thermocline and 200-1000 m resolution in the deep ocean. The horizontal resolution is 1°. The turbulent vertical mixing parameterization is based on the non-local K profile parameterization (KPP) of (Large et al., 1994). Open boundaries conditions are treated using a mixed active/passive scheme (Marchesiello et al., 2001). This scheme is used to force our regional configuration with monthly climatological large-scale boundary conditions from a ½° ORCA global ocean simulation (details available in Kessler and Gourdeau (2007)), while allowing anomalies to radiate out of the domain. The use of similar ROMS

configurations (e.g vertical resolution, mixed active/passive scheme, turbulent vertical mixing parameterization) in the WTSP is largely validated through studies demonstrating skills in simulating both the surface (Jullien et al., 2012, 2014; Marchesiello et al., 2010) and subsurface ocean circulation (Couvelard et al., 2008).

To compute the momentum and fresh water/heat fluxes, we also use a climatological forcing strategy. Indeed, documenting the inter-annual to decadal variability is beyond the scopes of our study, which justifies using climatological forcing fields. A monthly climatology of the momentum forcing is computed from the 1993-2013 period of the ERS1-2 scatterometer stress (http://cersat.ifremer.fr/oceanography-from-space/our-domains-of-research/air-sea-interaction/ers-ami-wind). Indeed, ERS derived forcing has been shown to produce adequate simulations of the Pacific Ocean dynamics (e.g, Cravatte et al., (2007)). A monthly climatology at 1/2° resolution computed from the Comprehensive Ocean–Atmosphere Data Set (COADS; Da Silva et al. 1994) is used for heat and fresh water forcing. In our set-up, ROMS also forces on line a biogeochemical model using a WENO5 advection scheme (i.e. five order weighted essentially non-oscillatory scheme; Shchepetkin and McWilliams, 1998). After a one year spin-up we stored 1-day averaged outputs for analysis. »

The configuration of the biogeochemical model is not well described. E.g., line 134: a modified version, which differs in the use of a full quota formation. How is it modified? How does it differ? 'variable' Redfield ratios. The Redfield ratio is always constant and always the same (i.e. the one that Redfield used). Replace by variable C:N:P (:Si : Fe:. . .?) ratios. Is the effect of N2 fixation (and denitrification) on alkalinity included in the model? This would be another biogeochemical impact of N2 fixation that should be reported.

This whole section, describing the PISCES-QUOTA model, has been revised. Careful attention has been paid not to refer to « variable Redfield ratio » and to stress differences between the PISCES common version and the quota version. To answer your particular question, $N_2$ fixation is indeed impacting alkalinity for both the implicit and explicit parameterization.

**« 2.1.2 PISCES**

In this study, we use a quota version of the standard PISCES model (Aumont and Bopp, 2006a; Aumont et al., 2015), which simulates the marine biological productivity and the biogeochemical cycles of carbon and the main biogenic elements and micronutrient (P, N, Si, Fe). This modified model, called PISCES-QUOTA, is extensively described in Kwiatkowski et al. (2018, in press). Our version is essentially identical to Kwiatkowski's version that included an additional picophytoplankton group, except that this latter group has been removed and replaced by the Trichodesmium compartment. Here we only highlight the main characteristics of the model and the

specifics of our model version. Our version of PISCES-QUOTA has then 39 prognostic compartments. As in the standard PISCES version, phytoplankton growth is limited by the availability of five nutrients: nitrate and ammonium, phosphate, silicate and iron. Five living compartments are represented: Three phytoplankton groups corresponding to nanophytoplankton, diatoms, and *Trichodesmium* and two zooplankton size-classes that are microzooplankton and mesozooplankton. The elemental composition of phytoplankton and non-living organic matter is variable and is prognostically predicted by the model. On the other hand, zooplankton is assumed to be strictly homeostatic, i.e. its stoichiometry is kept constant (e.g., Meunier et al., 2014; Sterner & Elser, 2002). Nutrients uptake as well as limitation of growth rate are modeled according to the chain model of Pahlow and Oschlies (2009). The P quota limits N assimilation which in turns limits phytoplankton growth. The phosphorus to nitrogen ratios of phytoplankton are described based on the potential allocation between P-rich biosynthesis machinery, N-rich light harvesting apparatus, a nutrient uptake component, the carbon storage, and the remainder (Daines et al., 2014; Klausmeier et al., 2004). This allocation depends on the cell size and on the environmental conditions.

Nutrients are delivered to the ocean through dust deposition, river runoff and mobilization from the sediment. The atmospheric deposition if iron is derived from a climatological dust simulation (Tegen and Fung, 1995). The iron from sediment is recognized as a significant source (Johnson et al., 1999; Moore et al., 2004). This iron source is indeed parameterized in PISCES as, basically, a time-constant flux of dissolved iron ($2 \ \mu mol.m^{-2}.day^{-1}$) applied over the whole sediment surface and modulated depending on depth. A detailed description of this sedimentary source is presented in Aumont et al. (2015). The initial conditions and biogeochemical fluxes (iron, phosphorus, nitrate, ...) at the boundaries of our domain are extracted from the World Ocean Atlas 2009 (https://www.nodc.noaa.gov/OC5/WOA09/woa09data.html). »

In addition to an improved description of N2 fixation, there should also be a description of the growth of diazotrophs as well as their loss terms (grazing, mortality,. . .) and the fate of the fixed N (loss to DOM? Lifetime?)

Indeed, the added section « Trichodesmium compartment » (already given at the beginning of this review) is describing the explicit simulation of *Trichodesmium* with information given, noticeably, on the grazing preferences of zooplankton groups towards *Trichodesmium*. The time-evolution equation of *Trichodesmium* biomass with the sources and sinks is also given in this section.

2.1.3 Trichodesmium compartment

For the purpose of this study, we implemented in the PISCES-QUOTA version an explicit representation of Trichodesmium. Therefore, as already stated, five living compartments are

modeled with three phytoplankton groups (nanophytoplankton, diatoms, and *Trichodesmium*) and two zooplankton groups (microzooplankton, and mesozooplankton). Similarly to nanophytoplankton (Equation 1 in Kwiatkowski et al., submitted), the equation of Trichodesmium evolution is computed as follows:

$$L_{Tri}^{T} = \frac{2,32.10^{-5} \times T^4 - 2,52.10^{-3} \times T^3 + 9,75.10^{-2} \times T^2 - 1,58 \times T + 9.12}{0.25}$$  (Eq. 1)

In this equation, $Tri_C$ is the carbon Trichodesmium biomass, and the seven terms on the right-hand side represent respectively growth, biosynthesis costs based on nitrate and ammonium, mortality, aggregation and grazing by micro- and mesozooplankton.

In our configuration, the photosynthesis growth rate of *Trichodesmium* is limited by light, temperature, phosphorus and iron availability. Photosynthesis growth rate of *Trichodesmium* ($\mu^{Tri}$) is computed as follows:  $\mu^{Tri} = \mu_{FixN_2} + \mu_{NO_3}^{Tri} + \mu_{NH_4}^{Tri}$  (Eq. 2)

where $\mu_{FixN2}$ denotes growth due to $N_2$ fixation, $\mu^{Tri}_{NO3}$ and $\mu^{Tri}_{NH4}$ represent growth sustained by $NO_3^-$ and $NH_4^+$ uptake, respectively. Moreover, a fraction of fixed nitrogen is released back to seawater, mainly as ammonia and dissolved organic nitrogen, by the simulated Trichodesmium compartment. Berthelot et al., (2015) estimated this fraction to be less than 10% when considering all diazotrophs. We set up this fraction at 5% of the total amount of fixed nitrogen.  For the other nutrients (i.e. iron and phosphorus), the same fraction is also released.

$N_2$ fixation is limited by the availability of phosphate, iron and light and is modulated by temperature.

Loss processes are natural mortality, and grazing by zooplankton. Natural mortality is considered to be similar to the other modeled phytoplankton species. Grazing on *Trichodesmium* is rarely described, but it is admitted that *Trichodesmium* represents a poor source of food for zooplankton (O'Neil and Romane, 1992) especially because they contain toxins (Hawser et al., 1992). On the other hand, many species of copepods have been shown to be able to graze on *Trichodesmium* despite the strong concentrations of toxins (O'Neil and Romane, 1992). For these reasons we applied two different coefficients for the grazing preference by mesozooplankton and microzooplankton (Table 1). For microzooplankton, grazing preference is halved to account for Trichodesmium toxicity, and for mesozooplankton the grazing preference is similar to that of the other phytoplankton species.The complete set of equations of *Trichodesmium* is detailed in Appendix 1. Table 1 presents the parameters that differ between Nanophytoplankton and Trichodesmium.

This setup reproduces $N_2$ fixation through an explicit representation of the Trichodesmium biomass (to be compared with often used implicit parameterizations (Assmann et al., 2010; Aumont et al., 2015; Dunne et al., 2013; Maier-Reimer et al., 2005; Zahariev et al., 2008) that links directly

environmental parameters to $N_2$ fixation without requiring the Trichodesmium biomass to be simulated).

line 163. Explain why 156E was chosen as western boundary of the test regions without sedimentary iron input? Doesn't this ensure that there is always iron being supplied from the western boundary of the Pacific Ocean?

We chose 156°E as the western boundary to remove the sedimentary iron source only in the south western Pacific islands, and to evaluate the impact of these islands and of iron from these islands on $N_2$ fixation.

**2 . The presentation of the results is often poor and not as convincing as is could and should be**

Part of this a language problem. Despite the impressive author list, no careful proof reading seems to have taken place before submission. There are many typos, incorrect words, wrong grammar and incomplete sentences. This can (and should) be improved. Some explanations are very vague and, at closer inspection, are not that convincing. For example, line 231/232: The bias 'beyond' (presumably 'eastward of'?) 170W is explained by a bias in iron concentrations, which, however occurs mostly west of 150W according to Fig.2.

The revised manuscript has undergone a thorough proof reading to look for typos and grammatical errors.  About your specific example, an undersestimation of the simulated iron concentration in TRI is displayed east of 170°W in figure 2 when compared to data from the 20°S transect (~0.2 in observations and ~0.4 in TRI simulation). The figure 9 strengthen the assumption that the iron biais is responsible to the N2 fixation biais in the south Pacific gyre. In addition we replaced 'beyond ' by 'eastward of'.

Fig. 4 Why show the vertical integral and the vertical average in separate panels?
The information looks very similar. Explain what differences the reader should see and understand.
We added text to explain the reason of these two integration layers:
« Some areas are sampled only in the surface layer (0-30m) while others have been sampled deeper. To overcome this sampling bias we compared the observations with N2 fixation rates simulated integrated over two different layers (0-30m and 0-150m). »

One motivation mentioned in the introduction was the comparison of biogeochemical controls and impacts between implicit and explicit representation of N2 fixation. The only comparisons shown are for surface chlorophyll (quite different) (Is the implicit diazotrophic biomass of the implicit representation included here?) and primary production (very similar). Both variables are

biogeochemically among the less relevant ones. Showing a comparison for N2 fixation rates, nutrient concentrations, export production, pCO2 and possibly oxygen would be much closer to the original goal of the paper. In my view, such a comparison is essential.

The main goal of this study is to evaluate the impact of explicit $N_2$ fixation on the tropical Pacific production (a change of the title of the study suggested by an anonymous reviewer now better reflect that main goal). However, we followed your recommendation to look at the $N_2$ fixation rates and carbon export in TRI and TRI_imp simulations.

Figure supp. 1 represents the carbon export (under the euphotic layer, in $\mu mol\ N.m^{-2}d^{-1}$ ) comparison and the figure supp. 2 represents the $N_2$ fixation rate comparison (integrated over top to 150m, panel a in mmol C. $s^{-2}.d^{-1}$ and panel b in percentage). We observe a carbon export greater in TRI simulation, the average across the Pacific of this difference is 0.1mmol $C.m^{-2}.d^{-1}$ or 4 %, and in LNLC regions the increase varies between 6 and 10 %. The $N_2$ fixation rates are greater in TRI simulation except in the warm pool, in the equatorial upwelling, and in Peru upwelling.

Fig. 2. Why does the run N2_imp have more chlorophyll along the eastern boundary and along the equator than run TRI? This is interesting and might point to some feedbacks in the system.

The nitrogen fixation rates

Indeed, this may be an indirect effect of the increased production of TRI (compared to N2_imp) notably within the gyres (Fig 2 & 10). This increased production drives a decrease in iron concentration within the euphotic zone in TRI (vs. N2_imp). Then, this negative iron anomaly (still compared to N2_imp) will, through the 3D circulation, impacts the sub-surface iron concentration. Then the water masses upwelled in the equatorial Pacific and along the eastern boundary have lower concentration in iron. Hence, less chl in TRI in these iron limited regions.

The comparison among modeled and measured iron concentrations in Fig.2 is very difficult to see. Try different figure types (larger blobs, overly observed 'blobs' on modeled map,...) Same for Fig.4

The goal of this comparison is to validate the spatial structure and the means. We have made the dots larger on the figure.

Fig. 9. Are currents on panels c and d different?

No, it's the same physical configuration, so the currents are identical. This is now acknowledged in the figure caption.

**3. minor points**

line 326 'cools temperature' is wrong either lowers temperature or cools the water.

Following your suggestion, we have modified the text.

line 349. What is meant by high islands?

They are the islands with high orography. We have modified the text.

Table 1 : Models parameters for Trichodemium and nanophytoplakton.

| Parameters | Symbol | Unity | Value | Reference |
|---|---|---|---|---|
| Maximum growth rate for Tricho. | $\mu^{Tri}_{max}$ | $d^{-1}$ | 0.25 | Breitbarth et al. (2007) |
| Maximum growth rate for Nano. | $\mu^{Nano}_{max}$ | $d^{-1}$ | 1.0 | |
| Initial slope P-I tricho | $\alpha I$ | $(W.m^{-2})^{-1} .d^{-1}$ | 0.072 | Breitbarth et al. (2008) and Hood et al. (2002) |
| Initial slope P-I nano | $\alpha I$ | $(W.m^{-2})^{-1} .d^{-1}$ | 2.0 | |
| Microzoo preference for Tricho. | pItri | - | 0.5 | |
| Microzoo preference for nano | pIP | - | 1.0 | |
| Maximum Fe/C in Tricho. | $\theta Fe,Trimax$ | $mol\ Fe.(mol\ C)^{-1}$ | $1.10^{-4}$ | Kustka et al. (2003) |
| Maximum Fe/C in nanophyto | $\theta Fe,Imax$ | $mol\ Fe.(mol\ C)^{-1}$ | $4.10^{-5}$ | |
| Maintenance iron | m | $mol\ Fe.(mol\ C)^{-1}$ | $1.4.10^{-5}$ | Kustka et al. (2003) |
| Marginal use efficiency | $\beta$ | $mol\ C.(mol\ Fe)^{-1}.day^{-1}$ | $1.4.10^{-4}$ | Kustka et al. (2003) |

Table 2 : List and description of the different experiments.

[revised manuscript text omitted]

Fig. Supp. 1 : Difference of fluxes of carbon export at euphotic layer between TRI simulation and TRI_imp simulation in mmol C.m$^{-2}$.d$^{-1}$ (a) and in percentage (b).

Fig. Supp. 2 : Depth-integrated (from 0 to 150m) rates of nitrogen fixation ($\mu$mol N.m$^{-2}$d$^{-1}$) in a) TRI simulation and b) TRI_imp simulation.

Fig. Supp. 3 : Physic balance of depth-integrated (from 0 to 150m) rate of iron concentrations (in mol Fe.m$^{-2}$.d$^{-1}$) averaged on south Pacific (red box, Fig. 1).

Fig. Supp. 4 : Seasonal cycle of Trichodesmium rate change (in mol C.m$^{-2}$.s$^{-1}$) in the South Pacific.

[Figure]

Figure 1

[Figure]

[Figure]

Figure 2

[Figure]

Figure 3

[Figure]

[Figure]

Figure 4

[Figure]

[Figure]

Figure 5

[Figure]

Figure 6

[Figure]

Figure 7

[Figure]

Figure 8

[Figure]

Figure 9

[Figure]

Figure 10

[Figure]

[Figure]

Figure Supp. 1

[Figure]

Figure Supp. 2

[Figure]

Figure Supp. 3

[Figure]

Figure Supp. 4